# Silencing cortical activity during sound-localization training impairs auditory perceptual learning

Victoria M. Bajo [1,5,6], Fernando R. Nodal [1,5], Clio Korn [1,3], Alexandra O. Constantinescu [1,4], Edward O. Mann [1], Edward S. Boyden III [2] & Andrew J. King [1,6]

The brain has a remarkable capacity to adapt to changes in sensory inputs and to learn from experience. However, the neural circuits responsible for this flexible processing remain poorly understood. Using optogenetic silencing of ArchT-expressing neurons in adult ferrets, we show that within-trial activity in primary auditory cortex (A1) is required for training-dependent recovery in sound-localization accuracy following monaural deprivation. Because localization accuracy under normal-hearing conditions was unaffected, this highlights a specific role for cortical activity in learning. A1-dependent plasticity appears to leave a memory trace that can be retrieved, facilitating adaptation during a second period of monaural deprivation. However, in ferrets in which learning was initially disrupted by perturbing A1 activity, subsequent optogenetic suppression during training no longer affected localization accuracy when one ear was occluded. After the initial learning phase, the reweighting of spatial cues that primarily underpins this plasticity may therefore occur in A1 target neurons.

[1] Department of Physiology, Anatomy and Genetics, University of Oxford, Parks Road, Oxford OX1 3PT, UK. [2] Departments of Biological Engineering and Brain and Cognitive Sciences, Massachusetts Institute of Technology, Cambridge, MA 02139, USA. [3] Present address: UCSF School of Medicine, San Francisco, CA 94143-0410, USA. [4] Present address: Institute of Cognitive Neuroscience, University College London, London WC1N 3AR, UK. [5] These authors contributed equally: Victoria M. Bajo, Fernando R. Nodal. [6] These authors jointly supervised this work: Victoria M. Bajo, Andrew J. King. Correspondence and requests for materials should be addressed to V.M.B. (email: victoria.bajo@dpag.ox.ac.uk) or to A.J.K. (email: andrew.king@dpag.ox.ac.uk)

Training can improve sensory skills in a range of tasks[1], and is useful for treating sensory disorders such as amblyopia[2] or hearing loss[3], as well for reversing age-related deficits in sensory processing[4]. Although plasticity of cortical function is generally considered to underpin perceptual learning, the processing stages involved remain controversial, with most of the evidence based on observation of physiological changes that correlate with improvements in performance[1,5]. Fewer attempts have been made to investigate the causal contribution of individual brain regions to perceptual learning. This is challenging to do because manipulating neural activity in sensory areas may affect task performance, making it difficult to identify a specific role for those areas in learning.

An ability to localize sounds accurately and rapidly is of great importance for the way in which humans and other species perceive and interact with their environment. Localization in the horizontal plane relies principally on a comparison of sound reaching the two ears and is therefore impaired immediately following hearing loss in one ear[6–8]. However, the auditory system can compensate for changes in the balance of inputs between the two ears, both during development[9–11] and in later life[6,8,12,13], and maintain accurate sound localization despite the presence of a hearing loss of up to 40–50 dB in one ear.

In mammals, adaptation to hearing loss in one ear is based principally on a reweighting of sound localization cues, with increased dependence on the monaural spectral cues provided by the normal-hearing ear[6,8,10,13], although compensatory adjustments in binaural sensitivity have also been observed[11,13–15]. Studies in adults have shown that adaptation depends on training[6,8,13] and is disrupted in animals in which the primary auditory cortex (A1)[16], and specifically layer V corticocollicular projection neurons[17], are lesioned. Although indicating that early auditory cortex is required for adaptation to an imbalance in inputs between the two ears, these approaches lack the temporal specificity necessary to determine at what stage in the learning process A1 is involved or the contribution of different types of cortical neurons. In particular, it is not known whether cortical activity following sound presentation during training is sufficient for adaptation to take place or whether A1 is also required for learning retrieval when abnormal spatial cues are experienced again.

In this study, we used reversible optogenetic silencing to demonstrate a causal link between A1 activity and adaptation to monaural deprivation by plugging one ear. We were able to dissociate the contribution of A1 to normal localization behavior and learning, and show that activity in its high-frequency region, where spatially informative spectral features are represented, is required when sounds are presented during training for adaptation to take place. Finally, our findings indicate that retrieval of the adaptive changes induced during learning can occur independently of A1, suggesting that these are likely to be consolidated in neural circuits to which this cortical area projects.

## Results

### Optogenetic silencing of neurons in the auditory cortex.
We trained adult ferrets to localize single bursts of noise that were presented from one of 12 loudspeakers positioned at 30° intervals in the horizontal plane (Fig. 1a). To examine the effects of perturbing A1 activity on the performance of the animals in this task, Archaerhodopsin T (ArchT) was expressed in A1 unilaterally by injecting AAV8/CAG-ArchT-GFP or AAV8/CaMKII-ArchT-GFP constructs in the dorsal part of the left middle ectosylvian gyrus (high-frequency A1) (Fig. 1b). We targeted the cortical region representing high sound frequencies (>8 kHz) because it is at these frequencies where direction-dependent spectral cues are

most prominent in ferrets and where cue reweighting has been shown to occur in monaurally occluded animals[10,13].

ArchT expression was visualized by co-expressed green flourescent protein (GFP), with the proportion of GFP neurons in the injected A1 (Fig. 2a, Fig. 3a, c) and presence of GFP axons and terminal fields in the contralateral auditory cortex (Fig. 2a, c) and in the auditory thalamus and midbrain (Fig. 2a, b, d, Fig. 3a, b, e, f) used to confirm the location and effectiveness of viral transfection. ArchT expressing cells were silenced by pairing pulses of green light ($\lambda$ = 532 nm) generated by a compact diode-pumped solid-state (DPSS) laser, delivered via an optical fiber implanted over the location of viral injections in A1, with sound stimulus presentation (Fig. 1a, b, Supplementary Fig. 1).

Suppression of cortical activity during light delivery was demonstrated electrophysiologically, both in vivo in ferrets with each viral construct (Fig. 2e–h, Supplementary Fig. 2) and in vitro in mice (Supplementary Fig. 3). Multiunit responses to auditory stimuli were recorded at 99 locations in the high-frequency region of ferret A1. Acoustically evoked activity was significantly reduced for units recorded within the region where viral injections had been made and where the fiber-optic cannula was within ~800 µm of the recording probe. No changes in the firing rate of cortical cells were found at distances greater than this or when the recording probe and/or the fiber-optic cannula were not located within the viral vector injection site. Activity suppression had a rapid onset and a slower recovery in vitro (up to 2 s after turning off the laser, Supplementary Fig. 3) than in vivo (Fig. 2 and Supplementary Fig. 2), where baseline activity was typically restored within 0.5 s, resembling the temporal dynamics of ArchT-mediated inactivation described in monkey cortex[18].

### Sound localization under normal hearing conditions.
As expected, the animals localized broadband noise more accurately than narrowband noise bursts, as shown by the higher proportion of correct responses (Fig. 1c; Wald $\chi^2_{df\ 1}$ = 1444.3, $P < 0.0001$) and the lower proportion of both front–back (Fig. 1d; Wald $\chi^2_{df\ 1}$ = 393.4, $P < 0.0001$) and left–right errors (Fig. 1e; Wald $\chi^2_{df\ 1}$ = 454.7, $P < 0.0001$). In keeping with previous studies[19], localization performance declined for both types of stimuli as the duration was reduced, resulting in fewer correct responses (Wald $\chi^2_{df\ 6}$ = 8241.3, $P < 0.0001$) and more front–back (Wald $\chi^2_{df\ 6}$ = 1042.3, $P < 0.0001$) and left–right errors (Wald $\chi^2_{df\ 6}$ = 403.6, $P < 0.0001$). During these sessions, green laser light pulses were delivered to the implant simultaneously with the auditory stimulus on 50% of randomly interleaved trials. No differences in sound localization accuracy (Wald $\chi^2_{df\ 1}$ = 0.176, $P = 0.675$), front–back errors (Wald $\chi^2_{df\ 1}$ = 0.468, $P = 0.494$), or left–right errors (Wald $\chi^2_{df\ 1}$ = 0.003, $P = 0.953$) were found when A1 activity was suppressed unilaterally (Fig. 1c–e), regardless of the location or duration of the sound stimulus. Response times were faster on correct trials (mean ± SD, 1.75 ± 0.37 s) than incorrect trials (1.96 ± 0.61 s) (analysis of the variance (ANOVA), $F_{1,46054}$ = 402.902, $P < 0.0001$), and were not affected by optogenetic suppression of A1 (ANOVA, $F_{1,46054}$ = 0.181, $P = 0.309$). Thus, disrupting activity in the high-frequency region of A1 during stimulus presentation had no apparent effect on auditory localization behavior under normal hearing conditions. The inter-trial interval was long enough (≥4–4.5 s) to ensure that any long lasting light-evoked change in neural activity (Fig. 2e–h) was over before the next stimulus presentation.

### Sound localization following monaural occlusion.
To investigate the role of A1 in adaptation to altered spatial cues, we compared the capacity of a group of control ferrets and the animals in which cortical activity was suppressed optogenetically

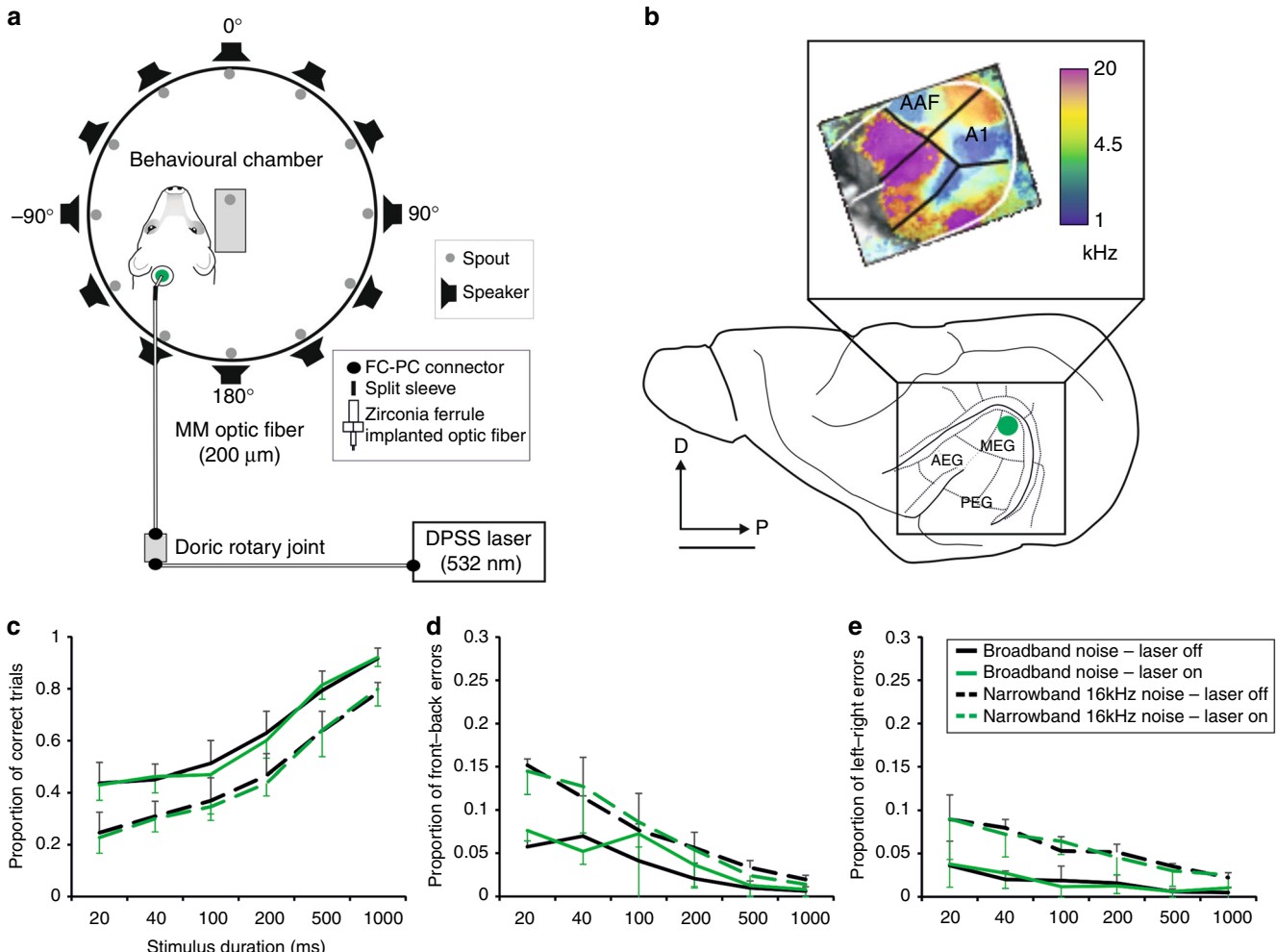

**Fig. 1** Effects of unilateral optogenetic suppression on sound localization accuracy. **a** Floor plan of the circular chamber used for behavioral testing, with 12 loudspeakers arranged at equal intervals around the periphery. The laser was located in a locked cupboard outside the chamber from where light was transmitted by a 2.0 m long fiber-optic cable (multi-mode fiber, core diameter 200 μm, 0.22 NA) connected via an FC connector to a second, 1.5 m long fiber-optic cable at a Doric rotary joint, which was attached to the ceiling of the chamber. During each testing session, the zirconia ferrule at the end of this second cable was connected by a split sleeve to an identical ferrule in the fiber-optic cannula implanted in the brain (Supplementary Fig. 1). The green dot indicates the location of the optical fiber implant in the high-frequency region of A1. **b** The ferret auditory cortex is located in the ectosylvian gyrus and comprises three areas: the anterior, middle, and posterior ectosylvian gyrus (AEG, MEG, and PEG, respectively). The primary auditory cortex (A1) and the anterior auditory field (AAF) are located in the MEG and are tonotopically organized, with neurons tuned to high frequencies (warm colors, inset) found in the dorsal corner of each field and neurons tuned to lower frequencies located more ventrally (cool colors). The inset shows the frequency map of one animal obtained using intrinsic optical imaging[66]. D, dorsal, P, posterior. Scale bar, 5 mm. **c–e** Sound localization accuracy was not affected by unilateral optogenetic suppression of A1 activity ($n = 13$). Two different sound stimuli, broadband noise and one octave narrowband noise centered at 16 kHz, were presented at six different durations (20–1000 ms) to examine the effects of suppression of activity in the high-frequency region of A1 in 50% of trials (randomly interleaved). **c** Proportion (mean ± s.d) of correct responses, **d** front–back errors, and **e** left–right errors. Source data are provided as a Source Data file

to recover sound localization accuracy during continuous monaural occlusion for 10 days. Although adaptation to unilateral hearing loss is seen at a range of stimulus durations[6,10,11], we used 1000-ms noise bursts for the earplugging experiments to be consistent with previous studies of cortical manipulations[16,17]. Importantly, those studies demonstrated very similar effects of monaural occlusion on the accuracy of both approach-to-target and head-orienting responses, which have a latency of ~200 ms, suggesting that the spatial cues provided by the onset of the target sound are used to drive the ferrets' adaptive localization behavior.

Consistent with the data shown in Fig. 1, the control group and the animals in which each stimulus presentation was paired with optogenetic silencing of A1 neurons localized the BBNs equally accurately in the session carried out prior to monaural occlusion

(Fig. 4a, b; preplug scores 0.94 ± 0.09 vs. 0.90 ± 0.06, respectively, $t$ test $t_{24} = 1.149$; $P = 0.262$). As soon as an earplug was inserted in the right ear, the proportion of correct responses fell to ~0.2 in all animals (Fig. 4b, c; controls 0.26 ± 0.09; ArchT 0.17 ± 0.05), confirming the importance of binaural cues for this task (Supplementary Fig. 4, Supplementary Fig. 5).

Daily sound localization training with the earplug in place throughout had a different effect on the localization accuracy of the two groups of animals. Control ferrets adapted with daily training, showing an almost complete recovery in the accuracy with which they localized BBNs over 10 days of continuous monaural deprivation (Fig. 4b–d black symbols and lines, Supplementary Fig. 4). This was manifest as an improvement in localization accuracy (slope significantly different from 0;

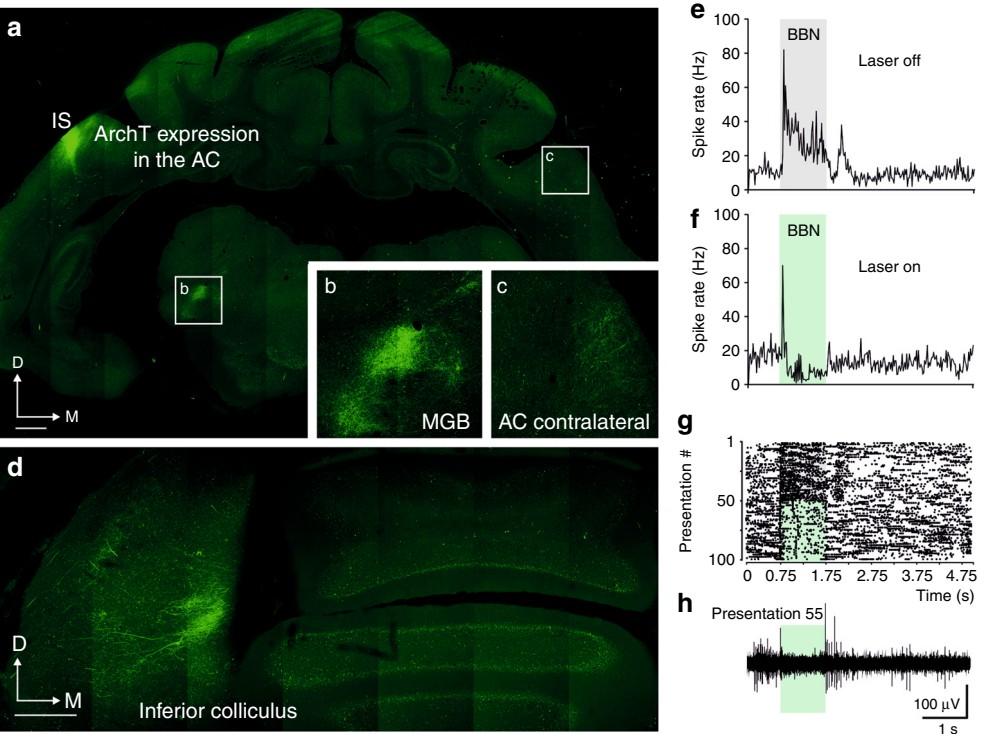

**Fig. 2** Optogenetic suppression of ferret A1. **a** ArchT-GFP expression in a coronal view of the auditory cortex and thalamus of a ferret brain (F1431) transfected with the AAV8-CaMKII-ArchT-GFP construct. IS injection site. **b, c** Higher magnification views of the corresponding boxed regions in (**a**). **d** GFP-labeled axons and terminals in the inferior colliculus originating from transfected neurons at the injection site shown in (**a**). **e–h** Neural activity recorded in a different animal (F1804) from a representative A1 unit expressing ArchT under the CaMKII promoter in response to 1000 ms bursts of broadband noise (BBN, gray bar) presented either alone (**e**) or paired with a 1000-ms light pulse (**f**) (green bar). (**e**) and (**f**) show 20 ms binned peristimulus time histograms. **g** The corresponding dot rasters for this unit grouped into laser-off (top half) and laser-on presentations (bottom half). **h** Activity recorded during a single presentation (#55 of the raster plot) with BBN paired with laser illumination (green) is shown in the bottom panel. Activation of ArchT by laser illumination resulted in suppression of the unit's sound-evoked response (t test, $t_4 = 6.584$, $P = 0.001$), including a loss of its offset response. Comparable spike rate to that found before BBN/laser illumination was observable as early as 0.5 s after laser offset (t test, $t_4 = 2.271$, $P = 0.086$). Auditory cortical cells continued expressing ArchT 30 months after the viral construct injections. Behavioral testing took place over a 30-month period and was stable throughout, with no impact of when the animals were tested on whether they adapted or not to monaural occlusion. Furthermore, once the brains had been processed histologically, subsequent neuronal quantification showed no differences in the overall number or density of transfected cells with survival time following viral vector injections. Scale bars in (**a**) and (**d**), 1 mm. AC auditory cortex, D dorsal, M medial, MGB medial geniculate body

ANOVA, $F_{1,128} = 204.534$, $P < 0.0001$) and a reduction in error magnitude on incorrect trials (slope significantly different from 0; ANOVA, $F_{1,128} = 10.014$, $P = 0.002$) with training.

A different result was found during optogenetic suppression of A1. Although they localized normally when intact binaural cues were available (Fig. 4a), suppression of cortical activity impaired the ability of these animals to adapt to a unilateral earplug (Fig. 4b–d, green symbols and lines, Supplementary Fig. 5; see Table 1 for detailed differences between groups). The proportion of correct responses increased (slope significantly different from 0; ANOVA, $F_{1,128} = 36.282$, $P < 0.0001$) (Fig. 4b, c) and the error magnitude on incorrect trials declined (slope significantly different from 0; ANOVA, $F_{1,128} = 15.636$, $P < 0.001$) (Fig. 4d) over the 10 days of monaural occlusion in the ArchT animals. However, while the errors decreased in size at the same rate in both groups (comparison of slopes: ANOVA, $F_{1,256} = 0.307$; $P = 0.580$), they were significantly larger throughout the period of monaural occlusion in the ArchT group (difference in error intercepts: ANOVA, $F_{1,257} = 173.886$; $P < 0.0001$) (Fig. 4d) and the localization accuracy of these animals recovered at a slower rate than the controls (ANOVA, $F_{4,580} = 5.863$, $P < 0.0001$) (Fig. 4b, c).

We can rule out the possibility of a nonspecific impairment because the control group included animals that received

injections of the same viral construct, but without the gene for ArchT, or were fitted with the same cranial implant with laser illumination but without a viral vector injection (Supplementary Table 1). Impaired adaptation during optogenetic suppression of A1 was observed with both CAG and CaMKII promotors, indicating that the activity of excitatory cortical pyramidal neurons is essential for learning to occur, without ruling out the possibility that the inhibitory neurons also play a role (Supplementary Fig. 5; adaptation slopes for CAG and CaMKII were not statistically different: ANOVA, $F_{1,126} = 1.354$, $P = 0.247$). In both control and experimental groups, the adaptation observed was based at least in part on the improvements in performance within each session as well as on the retention or consolidation of such learning between sessions. This is illustrated by the significant contribution of both the order of the trials within each testing session and the change in performance across training days to the general linear model (GLM) (Table 2).

All ferrets in which high-frequency A1 neurons were silenced unilaterally during stimulus presentation showed an immediate improvement in localization accuracy when the earplug was removed, indicating a dependence on binaural cues (Fig. 4b, Fig. 5, and Supplementary Fig. 5). However, the performance of these animals in the first post-plug session was much more

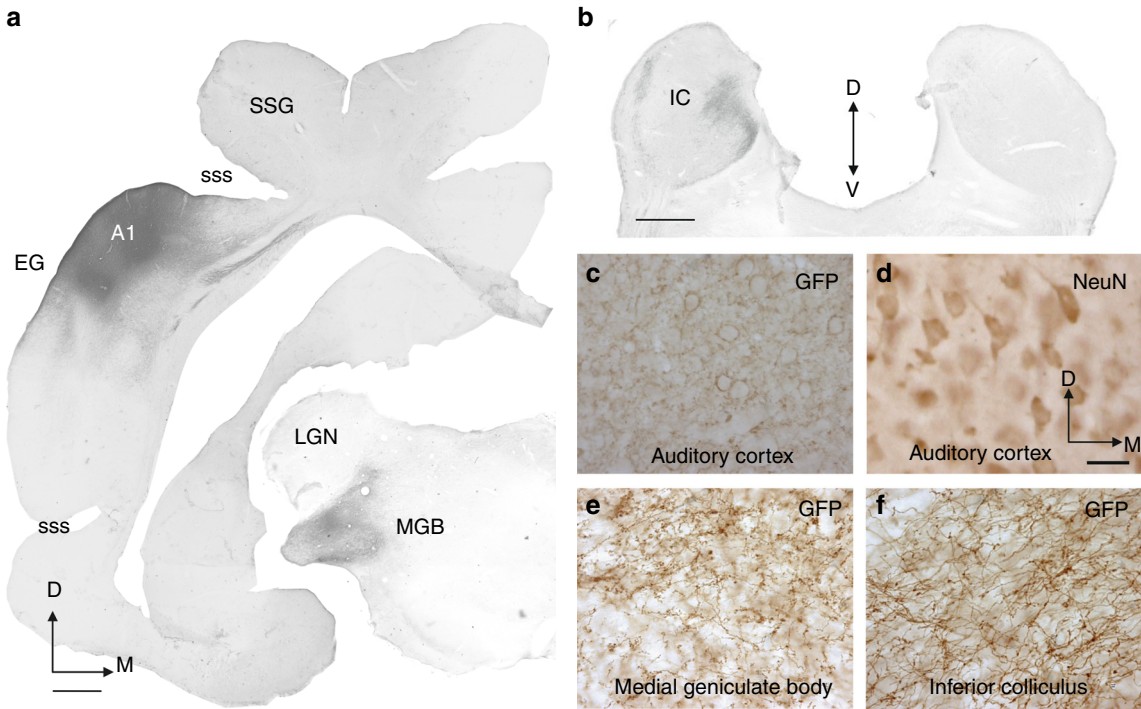

**Fig. 3** GFP Immunolabeling following transfection with the AAV8/CAG-ArchT-GFP construct. GFP staining (dark black/brown, shown for case F1115) was used as a reporter of ArchT expression. **a** GFP immunoreactivity following AAV8/CAG-ArchT-GFP injections in A1 and GFP-labeled axons and terminals in the medial geniculate body (MGB). **b** GFP-labeled axons and terminals in the inferior colliculus (IC). **c–f** High-power views. GFP-immunopositive (**c**) and NeuN-immunopositive (**d**) neurons at the location of the injection site in the cortex and, in the same animal, GFP labeling in fibers and terminals in the MGB (**e**) and IC (**f**). ArchT expression was identified on the basis of GFP immunoreactivity in the auditory cortex and compared with the distribution of NeuN-immunopositive neurons in order to estimate transfection rates for the two different constructs in ferrets used for behavioral testing (30 months after injections) and those used only for anatomy or recordings (4 weeks after injections). Stereological estimations made using the optical fractionator probe (n = 5 cases, 3 using CAG and 2 using CaMKII promoters) revealed that when AAV8/CAG-ArchT-GFP was used, 84.4 ± 14.8% (mean ± SD) of the neurons in the injection site were transfected, whereas, with AAV8/CaMKII-ArchT-GFP, the transfection rate was 60.03 ± 14.8% (mean ± SD). Other abbreviations: A1 primary auditory cortex, D dorsal, EG ectosylvian gyrus, LGN lateral geniculate nucleus, M medial, SSG suprasylvian gyrus, sss suprasylvian sulcus, V ventral. Calibration bars, 1 mm in (**a**) and (**b**), 50 μm in (**d**). Orientation and calibration bars in (**d**) apply to (**c**, **e**, **f**)

variable than that of the control group (Supplementary Fig. 6; equality of variance test $F_{12,12} = 9.803$, $P < 0.0001$), with some ferrets achieving their pre-plug scores while others performed at a much lower level (post-plug range 0.42–0.93 in the ArchT-laser-on group vs. 0.75–0.97 in the control group). Localization accuracy on the last day of monaural occlusion was a good predictor of the first post-plug score ($r = 0.68$ in the ArchT-laser-on group, $r = 0.55$ in the control group), suggesting that silencing high-frequency A1 neurons resulted in both incomplete and abnormal adaptation. This "aftereffect" quickly disappeared, however, since both control and ArchT ferrets achieved equally high pre-plug scores before the start of the next period of monaural occlusion (ANOVA $F_{4,54} = 1.959$; $P = 0.114$).

**Auditory spatial learning leave a memory trace**. Given that adaptation involves a change in the way auditory spatial cues are processed in the brain, we tested the hypothesis that auditory spatial learning leaves a memory trace that facilitates adaptation to a second period of monaural occlusion. This was confirmed by occluding the same ear for a second time in control animals that had previously adapted to the unilateral hearing loss. A much smaller initial deficit was observed when the ear was replugged (proportion of correct responses 0.47 ± 0.18) than when the animals first experienced an earplug (0.26 ± 0.09) (Fig. 6a). Furthermore, most of the control ferrets achieved their maximum score by ~day 5 and remained at around that level until the end

of the second period of monaural occlusion. Consequently, they achieved higher scores (Table 1) and the slope of the adaptation function was flatter (ANOVA $F_{4,580} = 5.863$, $P < 0.0001$, post hoc, $P = 0.013$) than when they were first trained with one ear occluded.

We also retested the ArchT animals that had previously shown impaired adaptation when cortical activity was suppressed. This time, however, they were tested without delivering light to the cortical implant during each stimulus presentation (Fig. 6b). Despite the restoration of cortical activity, these animals adapted at the same rate as that observed during the first period of monaural deprivation when the high-frequency region of A1 was inactivated, and significantly more slowly than the control animals during their first period of monaural occlusion (slope 0.028 vs. 0.057; ANOVA, $F_{4,580} = 5.863$, $P < 0.0001$, post hoc test $P < 0.001$). When compared with the second period of monaural occlusion in the control group, plugging one ear during this second training block produced a greater initial drop in localization performance in the ferrets expressing ArchT, and their overall performance was significantly worse (GLM, $P < 0.00001$, Table 1), with lower scores achieved by the end of the earplug training run (Fig. 6a, b). Thus, optogenetic suppression of cortical activity not only impairs auditory spatial learning, but also results in less effective adaptation when the active auditory cortex is subsequently challenged by monaural occlusion.

When we again tested the effects of optogenetic suppression on the ability of the ferrets expressing ArchT in high-frequency A1

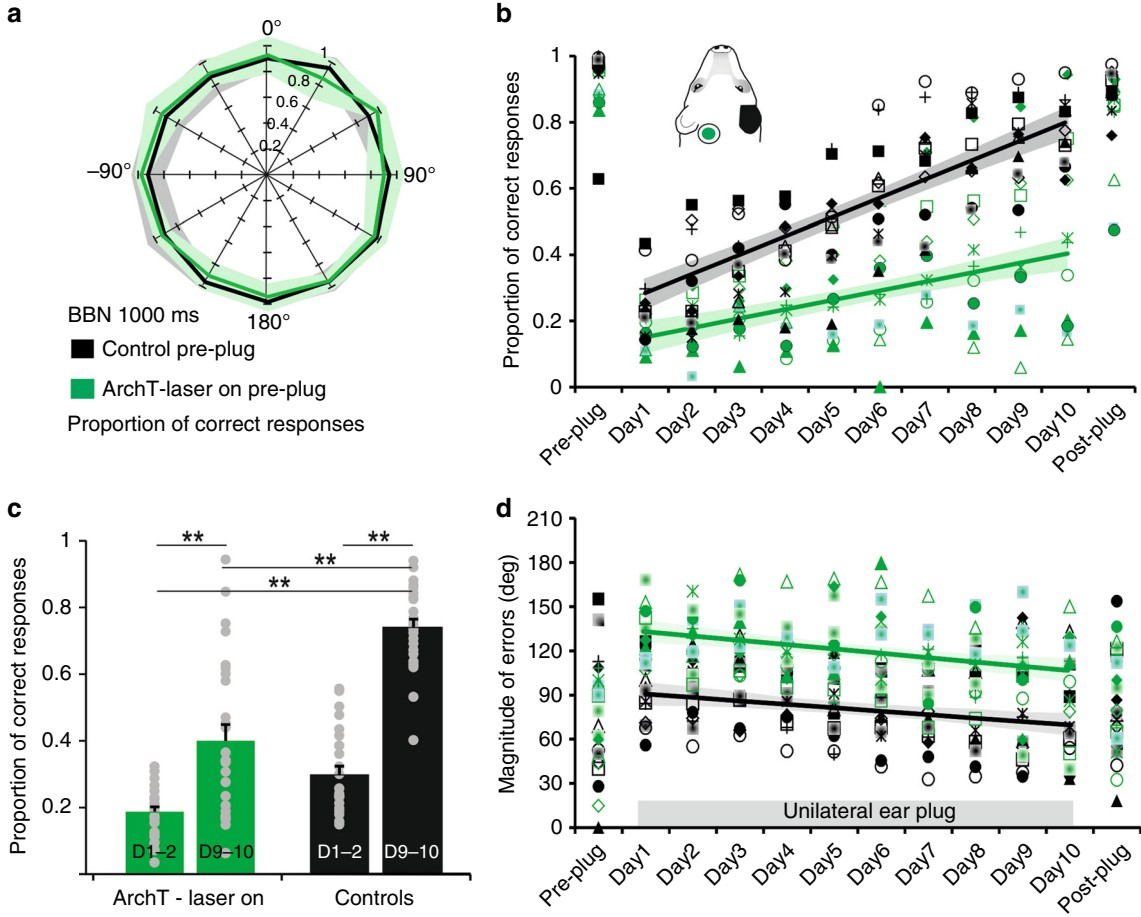

**Fig. 4** Optogenetic suppression of A1 impairs auditory spatial learning. **a** Sound localization in the session before the right ear was plugged shown by plotting the mean proportion of correct scores (and 95% confidence intervals) for 1000 ms bursts of broadband noise at each of 12 loudspeakers positioned at equal intervals in the horizontal plane (0° is directly in front). Data are from ferrets in which left A1 activity was suppressed by illumination of ArchT during each stimulus presentation (green, $n = 13$) and from a control group (black, $n = 13$). **b** Proportion of correct responses (averaged across all speaker locations) achieved by each animal in both groups in the Pre-plug session, on each of the 10 days over which the plug was worn (days 1–10), and in the session following its removal (Post-plug). **c** The difference in adaptation rate between these groups is shown by plotting the proportion of correct responses for the first 2 days and the last 2 days of monaural occlusion (ANOVA, $F_{3,100} = 62.531$, $P < 0.0001$, post hoc Scheffé test, ** indicates $P \leq 0.01$). **d** Magnitude of the localization errors on incorrect trials before, during and after this period of monaural occlusion. Symbols in (**b**) and (**d**) represent data from individual animals, and the lines and shaded areas are the best linear fits and 95% confidence intervals of these fits, respectively, over the 10 days of monaural occlusion. Source data are provided as a Source Data file

**Table 1 P values and odd ratios from the generalized linear model (GLM)**

|  | Control 1 | CAG Laser on 1 | CaMKII Laser on 1 | CAG Laser off | CaMKII Laser off | Control 2 | CAG Laser on 2 | CaMKII Laser on 2 |  |
|---|---|---|---|---|---|---|---|---|---|
| Control 1 |  | *0.563 | *0.463 | 1.506 | 1.403 | *3.537 | 1.381 | 1.155 |  |
| CAG Laser on 1 | *0.022 |  | 0.824 | *2.676 | *2.493 | *6.285 | *2.454 | 2.052 |  |
| CaMKII Laser on 1 | *0.004 | 0.517 |  | *3.249 | *3.028 | *7.631 | *2.980 | *2.491 |  |
| CAG Laser off | 0.103 | *0.000 | *0.000 |  | 0.932 | *2.349 | 0.917 | 0.767 | Odds ratio |
| CaMKII Laser off | 0.198 | *0.002 | *0.000 | 0.813 |  | *2.520 | 0.984 | 0.823 |  |
| Control 2 | *0.000 | *0.000 | *0.000 | *0.001 | *0.000 |  | 0.392 | 0.326 |  |
| CAG Laser on 2 | 0.199 | *0.000 | *0.000 | 0.388 | 0.958 | *0.000 |  | 0.836 |  |
| CaMKII Laser on 2 | 0.586 | *0.016 | *0.000 | 0.375 | 0.062 | *0.000 | 0.550 |  |  |
|  |  |  |  | *P* values |  |  |  |  |  |

GLM via PQL (penalized Quasi-Likelihood procedure) was performed using the *R* function glmmPQL. Significant values are shown by asterisks (*P* < 0.05). Control 1 and 2: control group data for the first and second periods of monaural occlusion, respectively; CAG and CaMKII: different promoters used in the ArchT group. Laser on 1: first period of monaural occlusion with the laser on; laser on 2: second period of monaural occlusion with the laser on

**Table 2 Odds ratios for between-session and within-session trial order**

| Odds ratio | Day of monaural occlusion | Trial order |
|---|---|---|
| Control 1 | *1.2596 | *1.0010 |
| CAG, Laser on 1 | *1.1527 | *1.0031 |
| CaMKII, Laser on 1 | *1.1665 | *1.0051 |
| CAG, Laser off | *1.1466 | *1.0036 |
| CaMKII, Laser off | *1.1223 | *1.0021 |
| Control 2 | *1.1769 | 0.9991 |
| CAG, Laser on 2 | *1.1325 | *1.0038 |
| CaMKII, Laser on 2 | *1.1586 | *1.0054 |

Asterisks indicate when the factors day of monaural occlusion or trial order within each session had a significant ($P < 0.05$) effect on localization performance. Generalized linear model via PQL (penalized Quasi-Likelihood procedure) performed using the *R* function glmmPQL. See Table 1 for definition of animal groups

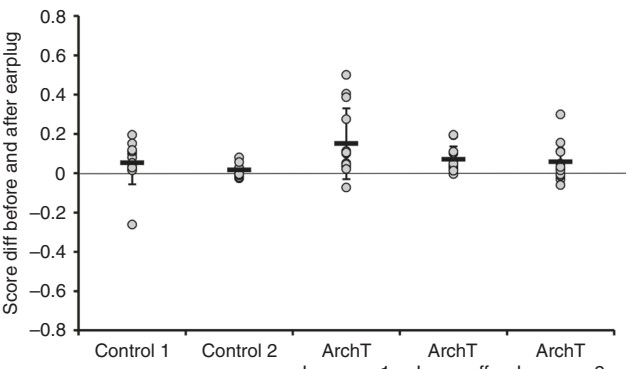

**Fig. 5** Difference between localization accuracy before and after monaural occlusion. The difference between the proportion of correct responses is shown for individual animals prior to plugging one ear and following removal of the earplug. The control group of ferrets underwent two periods of training (Control 1, $n = 13$, and Control 2, $n = 9$), whereas the animals in which ArchT was expressed in high-frequency A1 received three blocks of training whilst wearing an earplug, with laser light delivered to the implanted optical fiber on each trial during the first (ArchT laser on 1, $n = 13$) and third block (ArchT laser on 2, $n = 12$), but not during the second block (ArchT laser off, $n = 13$). The gray symbols represent data from individual animals, and the tick marks and error bars indicate the mean ± SD. Source data are provided as a Source Data file

to adapt to an earplug, we found that their performance changed in precisely the same fashion as in their previous training block without cortical inactivation (Fig. 6c) (slopes 0.030 vs. 0.028; ANOVA, $F_{4,580} = 5.863$, $P < 0.0001$, post hoc test $P = 1$; initial scores $0.361 ± 0.116$ vs. $0.376 ± 0.136$, final scores $0.616 ± 0.196$ vs. $0.603 ± 0.140$). Thus, in contrast to the first earplugging run with the laser on, the limited capacity of these animals to adapt to further periods of monaural occlusion no longer appears to be dependent on activity in high-frequency A1.

## Discussion

Our results show that transient perturbation of A1 activity initiated at the onset of the sound presentation during a localization training task impairs the ability of adult ferrets to adapt to altered spatial cues induced by reversible occlusion of one ear. Critically, auditory localization accuracy under normal-hearing conditions was unaffected during optogenetic suppression of cortical activity, demonstrating a specific role for A1 in learning. Further training of these animals without delivering

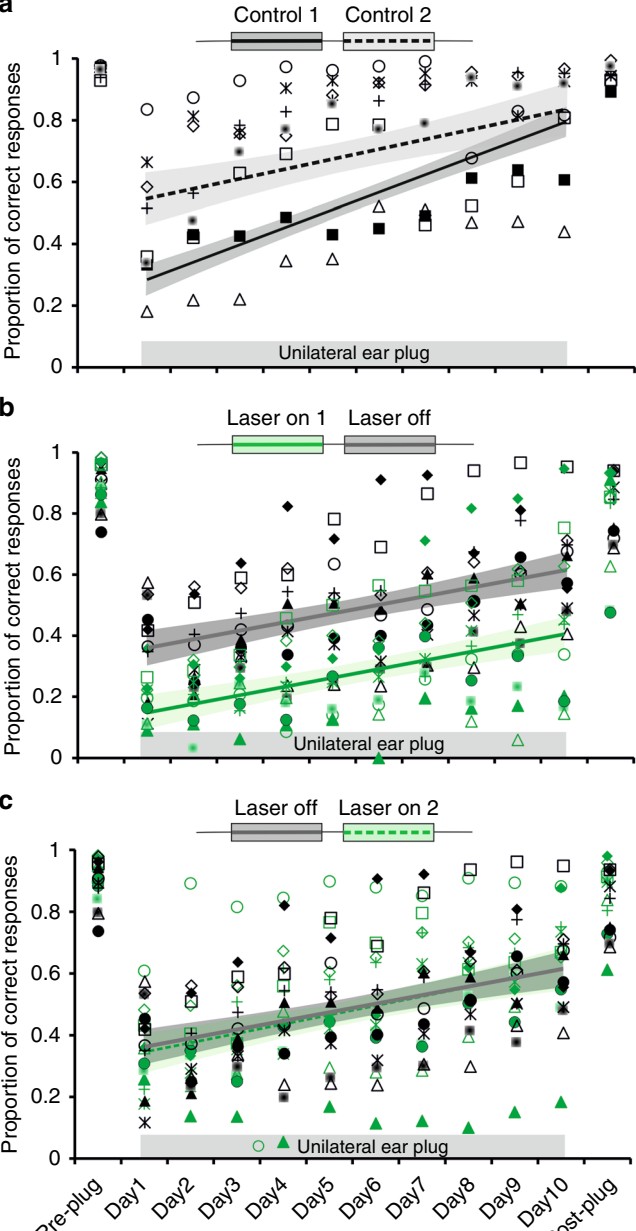

**Fig. 6** Learning retrieval during successive periods of monaural occlusion. Proportion of correct responses (averaged across all speaker locations) is shown in the session before the right ear was plugged (Pre-plug), on each of the 10 days over which the plug was worn (days 1–10), and in the session following its removal (Post-plug). **a** Data from the control group ($n = 13$) are shown for two 10-day periods of monaural occlusion: lines and shaded areas correspond to the best linear fits and 95% confidence intervals, respectively, while the symbols represent individual performance during the second period of monaural deprivation (individual animal data for the first period are shown in Fig. 4b). **b** Data from the ferrets in which ArchT was expressed in high-frequency A1 ($n = 13$) are shown for the first two 10-day periods of monaural occlusion, with light delivered to the implanted optical fiber to suppress cortical activity on every trial during the first block (laser on 1, green), but not during the second block (laser off, gray). Colored symbols represent individual performance under each condition. **c** Data from the ferrets in which ArchT was expressed in high-frequency A1 for the second and third 10-day periods of monaural occlusion; the laser was off during the second block (gray, as in **b**, $n = 13$), but on during each stimulus presentation in the third block (laser on 2, green, $n = 12$). Note that perturbing cortical activity no longer has any effect on the localization accuracy of the animals. Source data are provided as a Source Data file

light to the cranial implant produced less adaptation than in an equivalently trained control group and this limited improvement was unaffected by subsequent cortical inactivation. These findings suggest that although A1 is required for spatial learning when abnormal inputs are first experienced, the resulting auditory memory may reside elsewhere in the brain and that the extent of the adaptive plasticity achieved during the initial learning phase determines how much adaptation is possible with further training.

Previous studies in ferrets[20,21] and other species[22–26] have shown that unilateral lesions or inactivation of the auditory cortex produce contralateral localization deficits. This is the case, at least for brief sounds, even when A1 is specifically targeted using temporary inactivation methods[21,26]. Because a change in sound localization performance would complicate the interpretation of any effect on learning, we focussed on high-frequency A1 and showed that unilateral photostimulation of cortical neurons expressing the proton pump ArchT reduced spiking activity within ∼800 μm of the fiber-optic cannula and across much of the depth of the cortex. In contrast to the more extensive inactivation used in previous studies, this did not affect the accuracy with which ferrets localized either broadband or narrowband noise bursts at any of the sound durations tested, presumably because acoustically driven activity was still present in much of A1.

For the broadband stimuli used in the ear-plugging experiments, adaptation to monaural occlusion results principally from learning to up-weight the spectral localization cues available at the nonoccluded ear[6,8,13]. Ferrets raised with one ear occluded develop greater sensitivity to high-frequency spectral features that contain most of the directional information in the head-related transfer function[10,13]. Because adaptation in adult animals was greatly reduced by optogenetic suppression of activity in the region of A1 representing those frequencies, it is likely that the impairment in learning primarily reflects a reduced ability to reweight the different cues used for localizing sound. This reweighting has been shown to occur as soon as human listeners experience an earplug in one ear[7], which could therefore explain why our ArchT ferrets made larger errors than the controls from the first day of monaural occlusion onwards.

Nevertheless, removal of the earplug normally results in a small and transient aftereffect[6], indicating that adaptive changes in binaural sensitivity also take place[13]. We found that optogenetic suppression of A1 during monaural occlusion not only reduced the capacity of ferrets to learn to accommodate the altered cues, but also resulted in much greater variation in response accuracy immediately following earplug removal than in the control group. This suggests that perturbing A1 activity led to both incomplete and abnormal changes in cue processing, which recovered following further exposure to normal inputs.

In these experiments, the ferrets wore an earplug continuously for each 10-day period of monaural occlusion and therefore experienced asymmetric hearing loss both during the twice-daily training sessions and in their home cages. Improvements in performance occurred during the training sessions and were maintained between sessions. This is consistent with our previous observation that the rate and extent of adaptation increase with the frequency of training sessions[6]. However, in humans, adaptation to monaural occlusion does not occur if the training is compressed into one day, indicating that off-line memory consolidation is also required[8]. By taking advantage of the transient nature of optogenetic suppression with ArchT, we were able to show that activity in A1 during the sound-localization task is critical for learning, further highlighting the importance of behavioral training in promoting the recovery of performance accuracy in the presence of hearing loss in one ear.

Previous studies have shown that training-dependent improvements in sound detection or discrimination are associated with plasticity in the response properties of A1 neurons[27–31], which is likely driven by top-down inputs[31]. Furthermore, direct activation of sensory cortical areas can induce temporary behavioral improvements in the absence of training[32–34]. Cortical stimulation[35] or activation of neuromodulatory inputs[36–38] can also enhance perceptual learning, while suppression of A1 activity disrupts learning[39]. Of particular relevance to our results, auditory training has been shown to compensate for deleterious effects of altered experience during infancy on A1 response properties[40] and behavioral sound detection thresholds[41]. Our findings provide further evidence for the role of training in restoring auditory function and for a critical role for A1 in learning. More specifically, they demonstrate that for normal learning to take place, cortical activity is specifically required when auditory stimuli are presented during training and that intact cortical activity between training sessions is not sufficient. It has been argued that transient alterations in the balance of cortical excitation and inhibition contribute to learning and changes in cortical response properties[42]. We have shown that activity in excitatory neurons is required for adaptation to asymmetric hearing loss, but plasticity of inhibitory synapses is also likely to be involved, with these changes triggered by neuromodulatory inputs that likely transmit reinforcement signals to auditory cortex[43,44].

Although the plasticity in cue processing that underpins adaptation to asymmetric hearing loss is context-dependent and rapidly reversible[8,10], the behavioral consequences of a second period of monaural deprivation suggest that the previous learning leaves a memory trace that can be retrieved when the same abnormal hearing conditions are experienced again. Thus, in the control group, a much smaller initial deficit was induced by occluding the same ear for a second time, with correspondingly less adaptation required to approach pre-plug levels of localization accuracy. Similarly, the more limited adaptation observed during optogenetic suppression of high-frequency A1 seems to determine the extent of the plasticity induced by subsequent training with the laser off. This reduced capacity to accommodate abnormal inputs with further training was unaffected when cortical activity was again perturbed by turning the laser on, suggesting that retrieval of previous learning may not be dependent on A1. This could then explain why both control and ArchT animals show a comparable rate of adaptation, with the much higher scores achieved by the controls reflecting the greater learning that occurs during the first earplugging run with normal A1 activity.

Learning-related changes in the functional organization of A1 disappear with further training over time, even though improvements in behavioral performance are retained[36,45]. While they may still be manifest in A1 in other ways, such as via changes in dendritic spine dynamics[46,47], it is also possible that the neural substrate for learned perceptual abilities, or in the case of our results a previously learned ability to reweight different spatial cues, lies elsewhere in the brain. For instance, selective strengthening of auditory corticostriatal synapses has been demonstrated over the course of learning an auditory discrimination task[48], while higher-order auditory cortical areas have been implicated in the consolidation of fearful memories[49]. Other studies also suggest that distinct brain regions may contribute to initial and subsequent stages of learning[50,51].

An additional possibility is that A1 is required for learning and that the processing underpinning sound localization following the initial adaptation to monaural occlusion occurs at subcortical levels. This resembles the effects of lesions of the motor cortex, which prevent learning but not the execution of certain previously acquired motor skills that are instead thought to be mediated by

downstream circuits[52]. Indeed, there is growing evidence that behavioral training can trigger plasticity at subcortical sites within the auditory pathway[53–56], and that this may be enabled via descending projections[17,57–60]. Our results may, therefore, be consistent with the reverse hierarchical theory of perceptual learning, in which activity-dependent changes proceed in a top-down fashion from higher to lower levels of processing, where the relevant stimulus features may be represented more precisely[61–63].

## Methods

**Animals and viral vectors**. Thirty-one adult female ferrets (>6 months of age) and five C57BL/6 mice (3 months of age) were used in this study (Supplementary Table 1). All sample sizes were chosen based on previous behavioral and neurophysiological studies in our lab. Animals were assigned to different experimental groups by technicians who were blind to the purposes of the study. Portions of the data were also collected by individuals who were unaware of either the project aims or the experimental group. However, group allocation was necessarily known to the lead investigators throughout the study. Experiments were approved by the Committee on Animal Care and Ethical Review of the University of Oxford and performed under license from the UK Home Office in accordance with the Animal (Scientific Procedures) Act (1986, amended in 2012). Ferrets were housed in small groups (<8) within environmentally enriched laboratory cages. Mice were housed in standard laboratory cages.

Replication deficient AAV8/CAG-ArchT-GFP, AAV8/CaMKII-ArchT-GFP, or AAV8/CAG-GFP control viral constructs were obtained from the Synthetic Neurobiology Group, Media Lab, MIT, Cambridge, MA (ArchT plasmid and map available at Addgene 29777), and viral vectors were produced by the Vector Core Facility at the University of North Carolina at Chapel Hill, with titers of $10^{12}$ IU ml$^{-1}$. In ferrets, the virus was injected at 4 sites across the high frequency region of A1 (Figs. 1b, 2a, 3a, Supplementary Fig. 2a) at depths of 1.0 and 0.5 mm below the pial surface at each site. In one case (F1804) both constructs AAV8/CAG-ArchT-GFP and AAV8/CaMKII-ArchT-GFP were injected at different locations (four per construct) in A1 (Supplementary Fig. 2a).

The injections were performed using a pressure injector (Nanoejector, Drummond Instruments). The volume of each injection was 50.6 nL, made using glass micropipettes with a 15–20 µm tip (F1217 and F1411, and the first 19 cases in Supplementary Table 1). The pipette was left in place at least for 5 min following each injection. The post-injection survival time was up to 30 months in the ferrets used for behavior, and 4 weeks in the animals used for recordings and/or anatomy (F1115, F1431, F1003, F1110, F1120, and F1804; Supplementary Table 1). In all cases, injection sites and sizes were very similar, with comparable GFP-positive terminal fields in the MGB and IC (Fig. 2b, d and Supplementary Fig. 2c).

AAV8/CAG-ArchT-GFP and AAV8/CaMKII-ArchT-GFP were also injected in the motor cortex in five mice (0.5 µL of two lateral positions). Two of these animals were used for in vitro patch-clamping experiments 4 weeks later, one with each promoter (Supplementary Fig. 3), two were used as anatomical controls, and the last one used to confirm virus expression 8 months later. In each case, injections were made at different cortical depths in a single penetration.

**Behavioral testing**. Ferrets were trained by positive-operant conditioning in a horizontal circular arena (Ø 140 cm) surrounded by 12 loudspeakers, spaced 30° apart, and located within a soundproof chamber (Fig. 1). A platform at the center of the arena served as the start position for each trial; infrared beams at the front and back of the platform detected whether the ferret was in position (with its head oriented toward the loudspeaker at 0° and tail toward the loudspeaker at 180°). Trials were initiated when the ferret licked a spout at the end of the platform and maintained its position for 500–1000 ms. On 5% of trials, licking the central spout resulted in delivery of one water drop from this spout, which maintained the animal's motivation to initiate trials (these were not considered for analysis).

Following trial initiation, a single burst of noise was presented from one of the loudspeakers. The ferret had to approach the location of this sound in order to receive a calibrated reward of 0.15 ml of water from a spout located beneath the loudspeaker. Stimuli used were either broadband noise burst (BBN) with a low-pass cut-off frequency of 30 kHz or narrowband noise centered at 16 kHz with a width of one octave, generated afresh for each trial by Tucker-Davies Technologies (TDT) System III hardware (Tucker-Davis Technologies, Alachua, FL) and digitally flattened and matched for sound level across the 12 loudspeakers. Incorrect responses triggered repeated trials at the same location until the ferret performed correctly; these trials were also not considered during analysis. MATLAB (MathWorks Inc., Natick, MA) controlled the coordination of stimulus presentation and reward delivery and registered the first spout licked by the animal following each stimulus presentation.

During each testing session, the location of the sound was determined pseudorandomly, and the sound intensity was varied in 7 dB steps between 56 and 84 dB sound pressure level (SPL) to rule out the possibility that animals were using intensity cues to determine the sound's location. The duration of the sound was kept constant during each testing session. For the experiments that measured the

effects of optogenetic suppression of A1 on sound localization accuracy, the stimulus duration was gradually stepped down from 1000, 500, 200, 100, 40 to 20 ms after a total of ~300 trials had been performed at each duration. Experiments with narrowband noise always followed those with broadband noise. The earplugging experiments were carried out using BBN of 1000 ms in duration.

Experimental (ArchT) and control groups performed a similar number of sessions and trials. The ferrets were trained in blocks of 5 days for normal sound localization measurements and in blocks of 14 days for the earplugging experiments. During these periods, they were tested twice a day and the number of trials animals performed ranged from 50 to 100 trials per session. Each animal performed a minimum of 300 trials per sound duration under normal hearing conditions and a minimum of 150 trials per day when wearing an earplug. In a small percentage of cases (~15%) and especially in the first few days of monaural occlusion, an additional session was sometimes required to reach the minimum number of daily trials.

Activation of ArchT in vivo was achieved by illumination of the auditory cortex with green light delivered by an optical fiber. A compact DPSS laser (Shanghai Laser & Optics Century Co.) was used to generate green light ($\lambda = 532$ nm) with an intensity of 10 mW measured (Vector H410, Laser Physics UK) at the tip of the final optical fiber (irradiance ~3 mW/mm$^2$), which is sufficient to induce ArchT activation (~400 pA photocurrent) up to ~800–1000 µm from the fiber tip[64,65]. Modifications to the behavioral testing chamber enabled us to connect the laser to the implanted optic fiber without affecting the animals' ability to perform the behavioral task (Fig. 1a). The intensity at the tip of the final optical fiber was again measured after the animals were perfused and the brain was extracted from the skull (Supplementary Fig. 1) to check the permeability of the fiber and also its position.

Laser and sound presentation triggering were controlled by the same TDT hardware to ensure precise temporal congruency between them. Laser illumination was randomized within testing sessions with a probability of 0.5 during normal sound localization testing. However, for the earplugging experiments, laser illumination was either paired with sound delivery or not within each 14 days testing block, which included at least 10 days of continuous monaural occlusion. This was achieved by inserting a custom-fit foam plug (Earfit, Aearo, 3M) into the external auditory meatus of the right ear under medetomidine hydrochloride sedation (0.1 mg kg$^{-1}$ body weight i.m.). The earplug was held in place by filling the concha of the external ear with Otoform-K2 silicone impression material (Dreve Otoplastik, Unna, Germany). The health of the ear was checked before earplug insertion and following its removal by otoscopic examination and tympanometry (Kamplex KLT25 Audiometer, P.C. Werth). Acoustical measurements indicated that the earplugs produced 40–50 dB attenuation at frequencies >3.5 kHz, which rolled off gradually at lower frequencies[10].

**Surgery**. Ferrets were prepared for surgery 12 h before the procedure by pretreatment with the corticosteroid Solu-Medrone (methylprednisolone sodium succinate, 10 mg kg$^{-1}$, i.m., Pfizer) to prevent cerebral edema. Anesthesia was induced with Domitor (medetomidine hydrochloride, 0.022 mg/kg, i.m., Orion Pharma) and Ketaset (ketamine hydrochloride, 5 mg/kg, i.m., Fort Dodge Animal Health) in saline and maintained with IsoFlo (isoflurane, 1–3% with 100% oxygen as a carrier, Abbott Laboratories). At the beginning of surgery, animals were treated with an additional dose of Solu-Medrone, Antisedan (atipamezole hydrochloride, 0.06 mg kg$^{-1}$, s.c., Pfizer) to reverse the effects of Domitor, the antibiotic Synulox (clavulanate-potentiated amoxicillin, 0.1 mL kg$^{-1}$, s.c., Pfizer) to prevent infection, and atropine sulphate (0.06 mg kg$^{-1}$, s.c., Hameln Pharmaceuticals) to reduce pulmonary secretions. Analgesia was achieved with Vetergesic (buprenorphine hydrochloride, 0.03 mg kg$^{-1}$, s.c., Alstoe Animal Health) and Metacam (meloxicam, 0.2 mg kg$^{-1}$, s.c., Boehringer Ingelheim) during the surgery. Dopram (doxapram hydrochloride, 0.2 mg kg$^{-1}$, s.c., Norbrook Laboratories) was administered when necessary to prevent respiratory depression. Postoperative treatment included 1 postoperative dose of Solu-Medrone, 2 days of treatment with Vetergesic (0.1 mg kg$^{-1}$, s.c.), and 5 days of treatment with Metacam (2 mg kg$^{-1}$ orally) and Synulox (0.1 mL kg$^{-1}$, s.c.).

Ferrets were artificially ventilated after tracheal intubation, and respiratory rate, heart rate and $CO_2$ levels were monitored continuously. A rectal probe monitored the temperature of the animal, which was maintained using a BearHugger warming device at 38–39 °C. A solution of glucosate serum (0.9% sodium chloride, pH 7.2–7.4, plus 5% glucose) was delivered throughout the surgery at a rate of 3.0–5.0 mL h$^{-1}$ via a cannula inserted into a forelimb vein. During the surgery, the scalp was incised, the temporal muscles were resected and partially cut away above the area of the skull overlying the temporal lobe, a craniotomy was performed over the left auditory cortex, and the dura mater was punctured to allow free passage of the recording electrode and/or the glass micropipette into the neural tissue.

In the first implanted animals ($n = 2$), recordings were made at a number of sites in the auditory cortex to determine the location and borders of the A1 and its high frequency region. After viral vector injections, the implant was constructed by placing an optical fiber (multimode MM fiber, 200 µm diameter, Shanghai Laser & Optics Century Co.) in the left auditory cortex. The implant was positioned in the dorsal corner of the ectosylvian gyrus, which corresponds to the high-frequency region of A1 (Fig. 1b, Supplementary Fig. 2a); the locations of the recordings made during the injection surgeries and the landmarks of the ferret auditory cortex (sulci

and gyri positions) were used to guide the positioning of the implant. A plastic cap was placed around the portion of the implant extending above the skull and secured in place with bone cement. The scalp was then sutured around the implant. The implant was always recovered at the end of the behavior and after perfusion to test the optical permeability of the optical fiber by measuring the light intensity (Supplementary Fig. 1).

**Recordings in the ferret auditory cortex.** Four weeks after the initial surgery to inject the viral constructs, animals used for acute terminal recordings and those in which behavior testing had concluded (Supplementary Table 1) were prepared for surgery as stated before, except that anesthesia was maintained with an i.v. infusion of Domitor and Ketaset in saline throughout and not with Isoflo. The head was fixed in place by a metal plate cemented to the skull.

We recorded multi-unit activity using single-shank silicon probes (Neuronexus, Ann Arbor, MI), with 16 recording sites spaced at 150 μm intervals in 2 cases (F1421 and F1115), 2 shanks separated by 200 μm with recording sites spaced at 50 μm in 1 case (F1804; Supplementary Fig. 2), and four shanks separated by 500 μm with 32 recording sites spaced at 50 μm in four cases (Supplementary Table 1). Acoustic stimuli were generated using TDT system 3 hardware and presented via Panasonic earphone drivers (RPHV297) mounted on plastic otoscope specula with their output frequencies flattened (±5 dB) to ≤30 kHz. Closed-field calibrations of the sound-delivery system were performed using an 1/8th-in condenser microphone (Brüel and Kjær, Naerum, Denmark). Frequency–response areas of cortical neurons were constructed from the responses to pure-tone stimuli presented pseudorandomly at frequencies from 500 to 30 kHz, in one-third octave steps. Tones were 200 ms in duration (5 ms cosine ramped) and intensity levels were varied between 10 and 80 dB SPL in 10 dB increments. BBNs (40 Hz–30 kHz bandwidth and cosine ramped with a 10 ms rise/fall time; 30–80 dB SPL) of 200 ms or 1000 ms in duration were used.

Recording probe penetrations were made perpendicular to the cortical surface starting at the corner of the ectosylvian sulcus in close vicinity to the optical fiber, which was placed in the same position as it was in the chronic animals and connected to the same 532 nm DPSS laser activated using the same parameters (10 mW measured at the tip of the optical fiber; Supplementary Fig. 2a). Laser illumination was paired with 50% of the sound stimulus presentations, beginning at the same time. We then varied the positions of the recording probe and optical fiber, so that the distance between the two covered the same range as in previous studies[64,65].

**Histology.** Ferrets used exclusively for anatomy were perfused 4 weeks after making viral injections in the auditory cortex, while those used for acute electrophysiology (again 4 weeks after the injections) were perfused as soon as the recordings were finished. Implanted animals used for behavior were perfused 30 months after the viral vector injections were performed, with a subset of 4 used for recordings in the auditory cortex before perfusion (see Supplementary Table 1).

Animals were sedated with Domitor, overdosed with Euthatal (400 mg of pentobarbital sodium, i.p., Merial Animal Health), and perfused transcardially with 0.9% saline (weight/volume) and 4% paraformaldehyde (weight/volume) in 0.1 M phosphate buffer, pH 7.4. The brains were removed, embedded in sucrose, cut on a freezing microtome at a thickness of 45 μm, and collected in 4 sets of serial sections. The brains were usually cut in the coronal plane but to visualize all the injection sites together in the same section, one of them was cut in the parasagittal plane and the cortex was flattened before being cut.

Out of 21 ferret brains (13 ArchT cases with behavior, 2 with recordings only, 3 with anatomy, and plus 3 controls), 15 were used to visualize GFP fluorescence on the sections and 6 were used for permanent immunoreaction of GFP- and NeuN-positive cells using the DAB-avidin-biotin-complex (ABC) system (Fl and ABC, respectively; anatomy column in Supplementary Table 1). In the first group, one set of sections was Nissl stained with 0.5% cresyl violet to identify the limits of different structures in the ferret brain, a second set was mounted on gelatinized slides, dried, and coverslipped for examination under a fluorescence microscope, and the last two set of sections were incubated with rabbit GFP primary antibody followed by incubation in Alexa Fluor secondary antibody (anti-rabbit Alexa Fluor488 for GFP detection, Molecular Probes dilution 1:200; Fig. 2a–d). Using confocal microscopy (confocal LSM 710 Carl Zeiss Microimaging microscope), we verified that the location and size of the injections sites in the auditory cortex and the anterograde labeling of axons and terminal fields in the auditory thalamus and midbrain injections were comparable in all animals.

The second group (ABC) was used for stereological analysis of GFP-positive cells in the injection site in A1. The first and second sets of sections were respectively stained for Nissl substance and mounted on gelatinized slides, dried, and coverslipped. The other two sets of sections were immunoreacted for permanent staining using the ABC with either mouse anti-neural nuclei protein (NeuN) monoclonal antibody to label every neuron in the injection site or an anti-GFP mouse monoclonal antibody to label the neurons transfected with the viral construct and therefore expressing GFP. For these immunoreactions, sections were washed several times in 10 mM phosphate-buffered saline (PBS) with 0.4% Triton X$_{100}$ (PBS-Tx) to permeabilize the cell membranes and incubated in 5% normal horse serum in PB for 1 h to block unspecific staining. The sections were then incubated for 72 h at 5 °C in the primary antibody (mouse monoclonal anti-GFP,

dilution 1:400, Sigma-Aldrich or mouse monoclonal anti-NeuN, Map377, Millipore Corp.), rinsed several times in PBS, incubated with biotinylated horse anti-mouse IgG (H + L) (dilution 1:200, Vector Laboratories), rinsed again several times in PBS, incubated for 90 min in avidin-biotin peroxidase (Vectastain Elite ABC kit, Vector Labs), rinsed a final time in PBS, and then incubated with 0.4 mM 3,3'-diaminobenzidine (DAB) (Sigma-Aldrich) and 9.14 mM H$_2$O$_2$ in 0.1 M PB until the reaction product was visualized (Fig. 3).

StereoInvestigator Software (MBF Bioscience, MicroBrightField Inc., Williston, VT) was used for stereological estimations of NeuN and GFP neurons. The optical fractionator was chosen as the stereological probe with parameters set to obtain a coefficient of error, which represents the precision of the population size estimate, of <0.05 (optical dissector: 1/8th of the sections, grid size: 0.09 mm$^2$, counting frame: 100 × 100 μm) (Fig. 3c, d). From an initial group of six cases for stereology, we had to exclude one animal because no GFP expression was present (case F1217, see supplementary Table 1).

**Mouse experiments.** To validate in vitro the viral constructs used in ferrets and the efficiency of ArchT to hyperpolarize the transfected neurons, we injected the same constructs, AAV8/CAG-ArchT-GFP or AAV8/CamKII-ArchT-GFP, from the same lots in the motor cortex of five C57BL/6 mice (Supplementary Table 1). Animals were anesthetized with isoflurane (4% in O$_2$) with Marcaine as local anesthetic, and Vetergesic and Metacam were used for peri-operatory analgesia. One animal was initially injected with AAV8/CAG-ArchT-GFP and perfused 4 weeks later to validate the efficacy of transfection. Two mice were used to validate the absence of long-term abnormalities in the injected cortex and were perfused 9 months later following pentobarbital overdose (Supplementary Table 1). GFP fluorescent cells were found in the injected motor cortex in each of the three cases, with no differences between them in size or tissue appearance at the injection site. The last two cases (one with each construct injected) were used for in vitro patch-clamping experiments (Supplementary Fig. 3). Coronal slices (400 μm) of the motor cortex were prepared 4 weeks later in ACSF containing (in mM): 124 NaCl, 3 KCl, 1.25 KH$_2$PO$_4$, 5 MgSO$_4$, 3.4 CaCl$_2$, 26 NaHCO$_3$, and 10 glucose, pH 7.2–7.4, bubbled with carbogen gas (95% O$_2$/5% CO$_2$). Slices were immediately transferred to an interface chamber and maintained at room temperature for 1 h between humidified carbogen gas and ACSF.

Recordings were made at room temperature using a Multiclamp 700B amplifier (Axon Instruments, CA, USA) with a sampling rate of 10 kHz and digitized by pClamp software (Axon), and later analyzed off-line. Patch pipettes (3–5 MΩ) were pulled from standard-wall borosilicate capillaries (1.5 mm OD × 0.86 mm ID, Harvard Apparatus) and filled with intracellular solution (140 mm cesium gluconate, 1 mm KCl, 10 mm Hepes, 4 mm potassium phosphocreatine, 4 mm ATP-Mg and 0.4 mm GTP, and 5 mg/mL biocytin (pH adjusted to 7.3 with KOH; osmolarity, ~300 mosmol l⁻¹). Electrophysiological data were analyzed using custom-written routines in MATLAB. Membrane potential and interspike intervals were measured in 1-s time windows and compared to pre-illumination (baseline) values (ANOVA); the latency for recovery to baseline conditions was established when statistical differences were no longer detected. At the conclusion of the recordings, ArchT-transfected cells and biocytin-filled cells were identified using chicken anti-GFP (Molecular Probes A10262) and fluorescent-labeled secondary antibodies (Streptavidine, Alexa Fluor647 conjugate (S-21374) and Alexa Fluor488 goat anti-chicken antibody (A-11039)). Confocal images were captured using similar parameters of laser power, gain, pinhole and wavelengths with two channels assigned as the emission color; z-stacks were taken individually for each channel and then collapsed (Supplementary Fig. 3).

**Statistical methods.** Sound localization accuracy (proportion of correct scores) plus the incidence of front-back and left-right errors were quantified by submitting individual trials to univariate generalized linear models (bimodal distribution with probit link functions) for each parameter. Rate of adaptation during periods of monaural occlusion was quantified by fitting regression lines for individual animals to the proportion of correct responses on each day; 95% confidence intervals around the mean are shown by shaded areas in the figures. Comparison of the mean scores obtained at the start and end of each period of monaural occlusion and of the regression line slopes were made using ANOVA after the data were checked for normality by inspection of Q–Q plots and application of the Shapiro–Wilk test, with homoscedasticity tested by Levene's test of equality of variances.

**Reporting summary.** Further information on research design is available in the Nature Research Reporting Summary linked to this article.

## Data availability

The source data underlying Figs. 1c–e, 4–6 and Supplementary Figs. 3–6 are provided as a Source Data File. All relevant data are available on request from the authors.

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

## Acknowledgements

We thank the late Daniel Lunn from the Statistics Department for his advice and help on the implementation of the GLM, Amy Hammond-Kenny, Josh Gold, and Sue Spires for their help with behavioral testing, Louise Upton, Antonio Langfelder and Pantelis Antonoudiou for their contribution to the in vitro patching experiments, Fernando García-Moreno and Alan Wainman for help with the confocal imaging, Kit Reynolds for assistance with the anatomical experiments, Lily Goldblatt for her contribution to the recording analysis, and David Bannerman for comments on the manuscript. This work was supported by the Wellcome Trust (WT076508AIA, WT108369/Z/2015/Z Principal Research Fellowship to A.J.K.) and by Action on Hearing Loss (S72_Bajo to V.M.B. and A.J.K.).

## Author contributions

V.M.B., F.R.N., and A.J.K. conceived and designed the project; V.M.B. and F.R.N. performed research and analyzed all of the data; C.K. and A.O.C. contributed to the behavioral studies; E.S.B provided viral constructs and optogenetic support; E.O.M. performed in vitro recordings; V.M.B., F.R.N., and A.J.K. wrote the paper, with the assistance of E.O.M.; V.M.B. and A.J.K. supervised the work.

## Additional information

**Competing interests:** The authors declare no competing interests.

