## [Peer Review File · Nature Communications]

Reviewers' Comments:

Reviewer #1:

Remarks to the Author:

This is an elegant and well-designed study that examines the specific role of auditory cortical neurons in the ability of animals to adapt to altered auditory-spatial cues, so as to maintain good localization abilities based on the spatial cues now available. The key finding, that patterned cortical activity induced during training is a requirement of the learning of new spatial cues, is novel, and of broad significance to the field of neuroscience. I have several major comments that should be addressed, including the possibility that the effects observed by plugging and inactivation of auditory cortical neurons is non-specific to learning per se, and actually more related to processing of spatial cues. I am happy to be convinced otherwise, but would like to hear from the authors what their arguments are for this not being the case, given the data.

Major comments

The result that needs further explanation concerns the poorer localization abilities following removal of the plug. I don't understand the argument that interpreting the full set of localization cues normally when balanced binaural cues become available again. Doesn't this suggest that 'maintenance' of sensitivity to spatial cues—whichever available—is also a function of Arch-T expressing neurons? I cannot understand how this cannot be the case, given there is no a priori reason as to why normal binaural sensitivity should not have instantly been restored, as in the control animals. Why is this?

The possibility that A1 activity during the initial period of learning new spatial cues is the critical feature associated with the extent to which learning in the future occurs is intriguing, but raises the question as to why this should be the case? Why is this initial period so important? Is it not possible that some other interpretation can be called upon here, especially given the lower spatial abilities of the plugged/Arch-T deactivated group following removal of the plug. Doesn't this argue for a non-specific effect (with respect to learning) and for an effect of this procedure on spatial processing per se. this could be dependent on inputs from lower brain centres—those influenced by 'top-down' input from Arch-T sensitive neurons, but it seems to be the case that overall spatial processing is reduced in efficacy in the test group. Is this the case, and, if so, doesn't it suggest a non-specific (to learning) effect of the test paradigm?

Minor comments

Abstract:- Description of the key finding, the disruption of adaption to a second bout of auditory deprivation is reduced by optogenetic suppression of ArchT-expressing neurons, is rendered opaque in the abstract. Whilst the broader perspective of the study needs to be expressed in the abstract, understanding what experiment exactly was performed, and what was observed, particularly for the second aspect of the study, is not easy to glean.

Line 192 Missing 'with' twixt 'keeping' and 'the'. My preference would actually be to replace this with 'Consistent with the'

Lines 364-373. This paragraph reads more like a section from the Results. Please alter it to reflect the style of the rest of the Discussion.

Reviewer #2:

Remarks to the Author:

The goal of the paper is to investigate the role of the high frequency region of primary auditory cortex (A1) of ferrets in re-learning of sound localization following monaural deprivation by ear-plugging. The authors' conclusion is that the high-frequency area of primary auditory cortex is not required for normal sound localization of broadband stimuli but is involved in the learning of altered-spatial cues. Authors used virally delivered ArchT to suppress neuronal activity in A1. Ferrets were trained to localize broadband noise bursts and achieved a high performance on the behavior. Suppressing A1 did not affect performance in trained subjects, suggesting that the high frequency region of A1 is not required for normal sound localization performance. However, suppression of A1 over 10 days following monaural deprivation caused a deficit in the learning of the new sound localization cues, with the rate of relearning slowed down over 10 days of training as compared to controls. Re-plugging of the ear in the both groups for a second time provided mixed results.

This is a conceptually interesting study and the manuscript is a pleasure to read. These results provide a new perspective on the function of the auditory cortex in sound localization and habituation to new cues following sensory deprivation. Furthermore, using optogenetics in ferrets in auditory behavior is a major breakthrough.

1. My main concern is that a key piece of analysis in the manuscript provides somewhat unexpected results and is incomplete. In order to understand behavioral consequences of shining light on the cortex where neurons express ArchT, we need to understand the effects of the manipulation on neuronal activity. First off, there are very few examples provided (N=1 multi-unit in Figure 2B for ArchT under CamK2 promoter, and N=1 penetration in Sup Figure 2d and e for ArchT under CAG promoter for ferrets). Furthermore, these two examples show different response patterns. There is no statistics provided for the size of the effect, and the effects seem inconsistent with each other and with prior literature.

Figure 2B presents recordings from a multi-unit with and without laser (with ArchT delivered under CamK2 promoter). The authors claim that laser presentation reduces neuronal activity. Indeed, there is lower firing rate in middle as compared to top panel during the laser. However when the laser is turned off, there is no increase in neuronal activity in the middle panel (at 1-2 s post tone onset). This is consistent with another sample recording from supplementary figure 3f bottom panel, where there is a prolonged suppression of firing (albeit after a rebound spike which is not observed in Figure 2B), at a higher laser power. However, this result is at odds with the timecourse of effects reported in literature for ArchT which shows recovery <1s post laser onset (for example Han et al, *Frontiers*, 2011 Figure 2C; Johanssen et al., *PNAS*, 2014, Figure 1; also under CamK2 promoter).

Furthermore, it is unclear whether the data in the middle panel of 2B are taken from a different time period in the recording than the data in top panel, such that the reduction in firing rate could be due to any number of other effects, such as electrode drift, change in animal level of arousal, some other external signal. It would be helpful to see a more complete time course starting at 2 s pre tone/laser and ending when the activity returns to baseline (according to Sup Figure 3f, this would be 1-2 sec post laser offset) and demonstrate the time course of the reduction of the response. It would be informative to add results of laser-only (no tone) effects. It would also be reassuring to understand the effects across more units so that statistical analysis can be applied.

In Sup Figure 2, under CAG promoter, there seems to be no reduction in spontaneous firing rate, but in fact reduction of tone-evoked response only. This effect is different from Figure 2B. Oddly, in infragranular layers, the response seems to increase -- is this statistically significant? Also, the legend does not mention this -- "The reduction in firing ... in general, was less at deeper locations within A1." In fact, there seems to be an increase in responses at deeper locations. Furthermore, the lack of reduction of spontaneous activity seems at odds with the results in Sup Figure 3a, where there is a complete suppression of firing during laser presentation across different intensities -- one would

expect a corresponding reduction in spontaneous activity (analogous to the effects in sup 3f vs figure 2 for ArchT under CamK2 promoter).

2. The language in the paper that described the behavior is somewhat inconsistent. Adaptation and perceptual learning are used somewhat inter-changeably. I am not sure that either "adaptation" or "perceptual learning" is appropriate here, since the effects are so long-term. Perhaps it's "habituation" (see Kato et al., Neuron 2015). Similarly, the statement that this is a first study demonstrating the "causal role of A1" in learning is somewhat overstated -- there are some studies using cooling, as well as pharmacology or optogenetics (e.g. Aizenberg, NN 2013, Aizenberg, PLoS Biology, 2015) as well as the authors' own prior work.

3. For controls, two groups seem to be pulled together -- was there any difference between the groups?

4. The results presented in Figure 6 are difficult to interpret. When we compare the performance of Laser Off groups and Laser 2 groups to Control 2 group, what is different? Is final performance or slope different? Furthermore, in laser 1 group, would the same final performance level be reached if the animals were habituated for extended period of time? It also seems that the rate of learning is similar for Control 2, Laser 1, Laser Off and Laser 2 groups -- what is the mechanism that would be consistent with this pattern?

Minor comments:

1. Page 2 paragraph 3, insert "cues" into "monaural spectral cues" provided by the normal-hearing ear

2. Title of figure 3 -- "Immunolabeling of GFP in a ferret brain transfected with the AAV8/CAG-ArchT-GFP construct. GFP staining (dark black/brown) revealed using primary antibodies against GFP and the DAB-ABC detection kit was always and exclusively visible in cells expressing ArchT". Not sure whether that is what is shown in the figure. There is no stain of ArchT and GFP to show that GFP was always and exclusively visible in cell expressing ArchT. GFP expression is used to infer ArchT expression.

3. Page 8, first paragraph: "In keeping with the data shown in Fig. 1..."

4. Fig 4a, legend can be clarified -- wasn't there laser shined in controls as well? By Laser On, you mean ArchT-Laser on?

Reviewer #3:

Remarks to the Author:

Manuscript ID: NCOMMS-18-13273

Title: Silencing cortical activity during sound localization training prevents auditory perceptual learning

Authors: Victoria M. Bajo, Fernando R. Nodal, Clio Korn, Alexandra O. Constantinescu, Edward O. Mann, Edward S. Boyden, Andrew J. King

Comments to the Authors:

It is well documented that with appropriate training, animals and humans can adapt to unilateral hearing loss, and auditory cortical activity plays an important role in this process. The present study sought to determine whether auditory cortical activity is necessary during training (as opposed to simply throughout the period of unilateral adaptation) to promote sound localization recovery. The authors optogenetically silenced the high frequency region of left primary auditory cortex in adult ferrets, and tested their ability to adapt to a unilateral earplug. While optogenetic silencing had no effect on sound localization accuracy in normal-hearing animals, it impaired adaptation to the earplug. While well-written and clear, there are several aspects of the manuscript that could be strengthened. It is my hope that the comments below will be useful to the authors during this process.

Major:

1. More information should be provided for the behavioral data in Figure 4. Specifically, what kinds of errors did the animals make after earplugging, and how did the error profile change during adaptation? When they made errors, were the animals close (localizing the sounds to an adjacent stimulus location), and were there any differences between the groups? Addressing these questions would allow for a more nuanced interpretation of the data. For example, one possibility is that the poor adaptation in the optogenetic animals is simply explained by the fact that they started out with larger errors. If this were the case, we might expect to see that the error magnitude decreases at the same rate in both groups, but because the control group started with smaller errors, they are able to correct them earlier in training. Alternatively, both groups may display similar error magnitudes at the beginning of training, and the optogenetic silencing may actually impair the rate of error correction. One approach to help distinguish between these possibilities for the reader would be to show representative data for individual animals from each group at different stages of training in the form of stimulus location vs. response location plots (as in Kacelnik et al., 2006, Figure 1A).

2. The authors use the data presented in Figure 6a to conclude that "auditory spatial learning leaves a memory trace in the brain that facilitates adaptation to a second period of monaural occlusion." However, the data in Figure 6a appear to come from only 4 animals, and the effect in question looks to be largely driven by data from only one of those 4 animals. They go on to suggest that "optogenetic suppression of cortical activity ... results in less effective adaptation when the active auditory cortex is subsequently challenged by monaural occlusion" (lines 278-280) and they come to this conclusion by comparing the "Control 2" data in Figure 6a with the "Laser Off" data in Figure 6b. A larger sample size for the control group is needed before either of these conclusions can be confirmed.

3. For the sake of scientific transparency and replicability, more methodological details are needed about the virus injections. For example, what was the flow rate used for the injections? What equipment was used to perform the injections? How long were the pipettes or needles left in place after each injection before removing them? Similarly, important details are missing about the behavioral experiments. How many trials were presented in each session of the earplugging experiments? Did optogenetic and control groups experience the same number of trials over the same number of sessions? How often were the sessions? (Once every 24 hr? 48 hr?) These details are critical to convince the reader that the experimental and control groups received an equivalent degree of training, with similar opportunities for memory consolidation between training sessions.

Minor:

1. The title of the manuscript is a bit overstated. Silencing cortical activity does not prevent perceptual learning. Rather, silencing impairs or impedes learning. I suggest changing the title to more accurately reflect the reported findings.

2. I believe there is a missing word on line 57: "...increased dependence on the monaural spectral cues provided by the normal-hearing ear..."

3. "In particular, it is not known whether behavioral plasticity is driven by exposure to altered inputs throughout the period of sensory deprivation or specifically by the activity patterns induced by training." This statement (lines 65-67) is not quite true, and is self-contradictory. As the authors themselves point out, adaptation depends on training, suggesting that mere exposure to altered inputs is insufficient to drive behavioral plasticity. I suggest revising this statement, focusing more on the question of when auditory cortical activity is necessary for behavioral adaptation.

4. As seen in Figure 2b, and Supplementary Figure 3a and 3f, the effects of laser stimulation can

persist for up to several seconds, raising the possibility that cortical activity generally does not return to baseline before the next trial starts. What was the intertrial-interval during behavioral testing? Was it longer than the average length of time that the light-evoked effects persisted?

5. The opening sentence of the legend for Figure 3 is confusing. The figure itself does not attempt to demonstrate using immunohistochemistry that GFP and ArchT are co-localized. Co-localization is assumed from the viral construct. However, the opening sentence states that "Immunolabeling of GFP... was always and exclusively visible in cells expressing ArchT." I suggest revising this sentence.

6. More information should be provided for the stereological estimations reported in the legend of Figure 3. Do the numbers represent data pooled across animals used for behavioral testing and anatomy? If so, are there differences in the transfection estimates at 4 weeks post-injection, compared to 30 weeks post injection? How many animals contributed to each of these estimates?

7. "...[T]he learning deficit was not due to a differential effect on excitatory versus non-excitatory neurons..." This statement (lines 234-235) is not necessarily true. In order to determine whether there is or is not a differential effect of excitatory vs. inhibitory cells, one would need to use the CamKII promoter to target excitatory neurons (as the authors do here), use a different promoter to target inhibitory neurons (such as mDlx) and directly compare the effects. As the CAG promoter targets both excitatory and inhibitory neurons, and the effects were similar to the CamKII promoter, the most one can conclude is that excitatory neurons are involved. One cannot rule out the possibility that inhibitory neurons also play a role, as the authors themselves point out in the discussion (lines 371-372). I suggest revising the statement in question.

8. Table 1 does not provide information regarding the contribution of within-session improvement and/or between session consolidation to learning, yet is referenced as such in the text (line 241).

9. "...[P]erturbing A1 activity whenever sounds are presented as part of a localization training task impairs adaptation..." (lines 304-305). "...[W]ithin-trial activity in A1 is critical for learning..." (line 354). These statements do not address the fact that the effect of the optogenetic manipulation can persist well after sound offset, likely into the intertrial-interval. This caveat should be addressed in each of these instances, or the statements should be revised accordingly.

10. The discussion of the causal relationship between cortical activity and perceptual learning, and the role of training in restoring sensory function (lines 357-363) should be expanded. Both gain-of-function and loss-of-function studies have investigated the role of various brain regions (including auditory cortex) in perceptual learning, and in sensory restoration. See:

Gain of function and perceptual learning:

Glennon et al. (2018) Locus coeruleus activation accelerates perceptual learning. *Brain Res.* doi: 10.1016/j.brainres.2018.05.048.

Karim et al. (2006) Facilitating effect of 15-Hz repetitive transcranial magnetic stimulation on tactile perceptual learning. *J Cogn Neurosci* 18:1577-1585.

Pleger et al. (2006) Repetitive transcranial magnetic stimulation-induced changes in sensorimotor coupling parallel improvements of somatosensation in humans. *J Neurosci* 26:1945-1952.

Reed et al. (2011) Cortical map plasticity improves learning but is not necessary for improved performance. *Neuron* 70:121-131.

Shibata et al. (2011) Perceptual learning incepted by decoded fMRI neurofeedback without stimulus presentation. *Science* 334:1413-1415.

Tegenthoff et al. (2005) Improvement of tactile discrimination performance and enlargement of cortical somatosensory maps after 5 Hz rTMS. *PLoS Biol* 3:e362.

Loss of function and perceptual learning:

Caras and Sanes (2017) Top-down modulation of sensory cortex gates perceptual learning. *PNAS* 114:9972-9977.

Sensory restoration:

Kang R, Sarro EC, Sanes DH (2014) Auditory training during development mitigates a hearing loss-induced perceptual deficit. *Front Syst Neurosci* 8:49.

Voss et al. (2016) Pairing Cholinergic Enhancement with Perceptual Training Promotes Recovery of Age-Related Changes in Rat Primary Auditory Cortex. *Neural Plast* 2016:1801979.

11. While I appreciate the authors' attempt at transparency, it is often unclear which (and how many) animals were used for which experiments. For example, How many animals contributed to the data in Figure 1c-e? Did the animals that contributed data to Figure 1c-e also contribute data to Figure 4-6? I suggest adding some information to Supplementary Table 1 to make clear the specific figure panels to which each animal contributed. It would also be helpful to organize the table by experimental group, rather than by the animal identity. Finally, I suggest including sample sizes in every figure, either in the panels themselves, or in the figure legends.

Reviewers' comments:

Reviewer #1 (Remarks to the Author):

This is an elegant and well-designed study that examines the specific role of auditory cortical neurons in the ability of animals to adapt to altered auditory-spatial cues, so as to maintain good localization abilities based on the spatial cues now available. The key finding, that patterned cortical activity induced during training is a requirement of the learning of new spatial cues, is novel, and of broad significance to the field of neuroscience. I have several major comments that should be addressed, including the possibility that the effects observed by plugging and inactivation of auditory cortical neurons is non-specific to learning per se, and actually more related to processing of spatial cues. I am happy to be convinced otherwise, but would like to hear from the authors what their arguments are for this not being the case, given the data.

We are grateful for the reviewer's positive comments. A specific impairment in the processing of auditory spatial cues should result in an impairment in sound localization under normal hearing conditions. This was not observed in our data, as shown in Figure 1. No deficit was apparent during optogenetic perturbation of auditory cortical activity when either broadband or narrowband noise was used as a stimulus. Furthermore, localization accuracy was unchanged both when laser inactivation was interleaved with trials in which the cortex was not inactivated (data shown in Figure 1) and when the laser was switched on with the stimulus on every trial (pre-plug data, Figure 4a, b; Supplementary Figure 5, first row). An impairment in performance was observed only when the animals had to relearn to localize sound in the presence of altered spatial cues. As previously demonstrated by Kacelnik et al. (2006), a recovery in localization accuracy occurs during monaural occlusion only if modality-specific localization training is provided. While the control group in the present study exhibited the expected spatial learning, this was not the case in ferrets in which cortical activity was perturbed. We believe our results therefore do demonstrate a specific effect on auditory spatial learning (the question of what happens when the earplug is removed is addressed in the next paragraph).

Major comments

The result that needs further explanation concerns the poorer localization abilities following removal of the plug. I don't understand the argument that interpreting the full set of localization cues normally when balanced binaural cues become available again. Doesn't this suggest that 'maintenance' of sensitivity to spatial cues—whichever available—is also a function of Arch-T expressing neurons? I cannot understand how this cannot be the case, given there is no a priori reason as to why normal binaural sensitivity should not have instantly been restored, as in the control animals. Why is this?

We understand the point the reviewer is making. However, this assumes that sensitivity to binaural cues remains unchanged as a result of wearing an earplug for 10 days. The process of adapting to the altered spatial cues resulting from monaural occlusion is based principally on a change in the weighting of different spatial cues, i.e., localization accuracy recovers due to an increased dependence on the intact spectral localization cues provided by the non-occluded ear, as demonstrated by our previous work in ferrets experiencing monaural occlusion in infancy (Keating et al., 2013) or adulthood (Kacelnik et al., 2006). Furthermore, the same result was obtained in adult human participants who adapted with training to an earplug in one ear (Kumpik et al., 2010; Keating et al., 2016). Adaptation based solely on a context-dependent reweighting of different sound localization cues would then be expected to lead to an immediate return to pre-plug localization performance when normal binaural cues are restored by removal of the earplug.

However, we have also previously shown that adaptation to monaural occlusion can arise from a systematic shift in sensitivity to binaural cues (Keating et al., 2015, 2016). If those cues contribute to the localization behaviour of the animals, we would predict that restoration of a normal binaural input should produce an aftereffect toward the side of the previously plugged ear. Indeed, this is exactly what we found in our original study in ferrets (Figure 5 in Kacelnik et al. 2006). After the earplug was removed from ferrets that had been trained to adapt to altered binaural cues, the animals in that study exhibited a small but significant bias toward the side of the previously plugged ear. However, this aftereffect was transient, usually disappearing after the first session following earplug removal, and was much smaller than the bias in the opposite direction produced by the introduction of the earplug. In other words, binaural plasticity appears to be small and secondary to cue reweighting as the basis for the recovery in sound localization accuracy.

We also found that nearly all control ferrets in the present study performed slightly worse in the first post-plug session than in their pre-plug session, especially after the first period of monaural occlusion ('Control 1'; Figure 5; individual data in Supplementary Figure 4), and that their performance had recovered by the start of the next run a few days later (pre-plug values in 'Control 2', Figure 6). A larger overall difference between pre-plug and post-plug performance was found in the 'ArchT laser on' animals (ArchT laser on 1; Figure 5; Supplementary Figure 5; Supplementary Figure 6, which shows that the mutual information between response and stimulus location is lower in the post-plug session in these animals than in the controls). This is mainly due to greater variability in the performance of the ferrets in which A1 activity had been perturbed, with some animals localizing the auditory targets almost as accurately as the controls immediately earplug removal, while others made both fewer correct responses and larger errors (supplementary Figures 5 and 6). Interestingly, we found an overall greater bias toward the side of the previously occluded ear in ArchT animals (indicated by the points above $x = y$ in supplementary Figure 6b), perhaps implying that perturbing cortical activity produced abnormal changes in binaural sensitivity during localization training with one ear occluded. However, as with the controls, these effects were transient and by the start of each of the following runs (pre-plug values in 'Laser off' and in 'Laser on 2', Figure 6), all the ArchT animals were localizing accurately under normal hearing conditions.

Given the variability in the performance of different 'ArchT laser on' animals in the immediate post-plug session, we do not wish to make too much of this. This temporary impairment in auditory localization under normal hearing conditions is only observed in some animals and is also observed to a lesser degree in the control group. We therefore think this rules out reviewer 1's suggestion that 'maintenance' of sensitivity to spatial cues depends on the silenced neurons. Instead, we believe the data are indicative of abnormal plasticity in response to training with altered spatial cues, which is manifest both as a lower rate of learning during the period of monaural occlusion and a temporary failure (in some cases) to make sense of the restored relationship between the different spatial cues.

We have revised the text on page 6 (last paragraph) of the Results as follows (new text in red):

“All ferrets in which high-frequency A1 neurons were silenced unilaterally during stimulus presentation showed an immediate improvement in localization accuracy when the earplug was removed, indicating a dependence on binaural cues (Fig. 4b, Fig. 5 and **Supplementary Figure 5**). However, the performance of these animals in the first post-plug session was much more variable than that of the control group (Supplementary Figure 6; equality of variance F-test_(12,12) = 9.803, $P < 0.0001$), with some ferrets achieving their pre-plug scores while others performed at a much lower level (post-plug range **0.42-0.93** in the ArchT-laser-on group versus **0.75-0.97** in the control group). Localization accuracy on the

last day of monaural occlusion was a good predictor of the **first** post-plug score ($r = 0.68$ in the ArchT-laser-on group, $r = 0.55$ in the control group), suggesting that silencing high-frequency A1 neurons resulted **in both incomplete and abnormal adaptation. This ‘aftereffect’ quickly disappeared, however, since both control and ArchT ferrets achieved equally high pre-plug scores before the start of the next period of monaural occlusion (ANOVA $F_{(4,54)} = 1.959$; $P = 0.114$).**”

We have revised the Discussion on pages 8-9:

“For the broadband stimuli used in the ear-plugging experiments, adaptation to monaural occlusion results principally from learning to up-weight the spectral localization cues available at the non-occluded ear^{6,8,13}. **Ferrets raised with one ear occluded develop greater sensitivity to high-frequency spectral features that contain most of the directional information in the head-related transfer function^{10,13}.** Because adaptation in adult animals was greatly reduced by optogenetic suppression of activity in the region of A1 representing those frequencies, **it is likely** that the impairment in learning primarily reflects a reduced ability to reweight the different cues used for localizing sound. **Nevertheless, removal of the earplug normally results in a small and transient aftereffect⁶, indicating that adaptive changes in binaural sensitivity also take place¹³.** We found that optogenetic suppression of A1 during monaural occlusion not only reduced the capacity of ferrets to learn to accommodate the altered cues, but also resulted in much greater variation in response accuracy **immediately** following earplug removal than in the control group. **This suggests that perturbing A1 activity led to both incomplete and abnormal changes in cue processing, which recovered following further exposure to normal inputs.**”

The possibility that A1 activity during the initial period of learning new spatial cues is the critical feature associated with the extent to which learning in the future occurs is intriguing, but raises the question as to why this should be the case? Why is this initial period so important? Is it not possible that some other interpretation can be called upon here, especially given the lower spatial abilities of the plugged/Arch-T deactivated group following removal of the plug. Doesn't this argue for a non-specific effect (with respect to learning) and for an effect of this procedure on spatial processing per se. this could be dependent on inputs from lower brain centres—those influenced by ‘top-down’ input from Arch-T sensitive neurons, but it seems to be the case that overall spatial processing is reduced in efficacy in the test group. Is this the case, and, if so, doesn't it suggest a non-specific (to learning) effect of the test paradigm?

Our answer to the reviewer's previous point largely addresses this. The only time we observed a deficit in the ‘ArchT laser on’ group under normal hearing conditions was immediately after the earplug was removed (and even then, only in some of the ferrets). This is a transient effect and all the other data (Figure 1, plus all the pre-plug data) show no deficit in spatial processing when cortical activity is perturbed in the absence of an earplug. A difference from the controls is, however, consistently seen during learning to adapt to altered spatial cues, and – in some cases – very transiently when the relationship between different cues is changed again by removing the earplug.

We agree that ‘top-down’ input from ArchT-expressing neurons may be triggering changes in lower brain centres, and address this in the last paragraph of the Discussion. Our results indicate that A1 activity is critical during the initial phase of learning new cues, but less so when animals again experience the altered cues following re-exposure to normal inputs. This appears to be consistent with the reverse hierarchical theory of perceptual learning (Ahissar and Hochstein 2004), in which activity-dependent changes proceed in a top-down fashion from higher to lower levels of processing. Similarly, distinct brain areas are engaged during

different stages of other forms of learning (Cammarota et al. 2005; Mukherjee and Caroni, 2018).

We have expanded the last two paragraphs of the Discussion to address this point more directly (pages 10-11):

“Learning-related changes in the functional organization of A1 disappear with further training over time, even though improvements in behavioral performance are retained^{36,45}. While they may still be manifest in A1 in other ways, such as via changes in dendritic spine dynamics^{46,47}, it is also possible that the neural substrate for learned perceptual abilities, or in the case of our results a previously learned ability to reweight different spatial cues, lies elsewhere in the brain. For instance, selective strengthening of auditory corticostriatal synapses has been demonstrated over the course of learning an auditory discrimination task⁴⁸, while higher-order auditory cortical areas have been implicated in the consolidation of fearful memories⁴⁹. **Other studies have also implicated distinct brain regions in initial and subsequent stages of learning^{50,51}.**

An additional possibility is that A1 is required for learning and that the processing underpinning sound localization following the initial adaptation to monaural occlusion occurs at subcortical levels. This resembles the effects of lesions of the motor cortex, which prevent learning but not the execution of certain previously acquired motor skills that are instead thought to be mediated by downstream circuits⁵². Indeed, there is growing evidence that behavioral training can trigger plasticity at subcortical sites within the auditory pathway⁵³⁻⁵⁶, and that this may be enabled via descending projections^{17, 57-60}. Our results may therefore be consistent **with the reverse hierarchical theory of perceptual learning, in which activity-dependent changes proceed in a top-down fashion from higher to lower levels of processing, where the relevant stimulus features may be represented more precisely⁶¹⁻⁶³.**”

Minor comments

Abstract:- Description of the key finding, the disruption of adaption to a second bout of auditory deprivation is reduced by optogenetic suppression of ArchT-expressing neurons, is rendered opaque in the abstract. Whilst the broader perspective of the study needs to be expressed in the abstract, understanding what experiment exactly was performed, and what was observed, particularly for the second aspect of the study, is not easy to glean.

The sentence about the second round of adaptation with optogenetic suppression has been changed, and is hopefully now clearer. It now reads:

“However, in ferrets in which learning was initially disrupted by perturbing A1 activity, subsequent optogenetic suppression during training no longer altered localization accuracy when one ear was occluded.”

Line 192 Missing ‘with’ twixt ‘keeping’ and ‘the’. My preference would actually be to replace this with ‘Consistent with the’

We have changed the text as suggested (now page 5, paragraph 3).

Lines 364-373. This paragraph reads more like a section from the Results. Please alter it to reflect the style of the rest of the Discussion.

Changed as requested. The paragraph now reads (now end of page 9, top of page 10):

“It has been argued that transient alterations in the balance of cortical excitation and inhibition contribute to learning and changes in cortical response properties⁴². We have shown that activity in excitatory neurons is required for adaptation to asymmetric hearing loss, but plasticity of inhibitory synapses is also likely to be involved, with these changes triggered by neuromodulatory inputs that likely transmit reinforcement signals to auditory cortex^{43,44}”.

Reviewer #2 (Remarks to the Author):

The goal of the paper is to investigate the role of the high frequency region of primary auditory cortex (A1) of ferrets in re-learning of sound localization following monaural deprivation by ear-plugging. The authors' conclusion is that the high-frequency area of primary auditory cortex is not required for normal sound localization of broadband stimuli but is involved in the learning of altered-spatial cues. Authors used virally delivered ArchT to suppress neuronal activity in A1. Ferrets were trained to localize broadband noise bursts and achieved a high performance on the behavior. Suppressing A1 did not affect performance in trained subjects, suggesting that the high frequency region of A1 is not required for normal sound localization performance. However, suppression of A1 over 10 days following monaural deprivation caused a deficit in the learning of the new sound localization cues, with the rate of relearning slowed down over 10 days of training as compared to controls. Re-plugging of the ear in the both groups for a second time provided mixed results.

This is a conceptually interesting study and the manuscript is a pleasure to read. These results provide a new perspective on the function of the auditory cortex in sound localization and habituation to new cues following sensory deprivation. Furthermore, using optogenetics in ferrets in auditory behavior is a major breakthrough.

1. My main concern is that a key piece of analysis in the manuscript provides somewhat unexpected results and is incomplete. In order to understand behavioral consequences of shining light on the cortex where neurons express ArchT, we need to understand the effects of the manipulation on neuronal activity. First off, there are very few examples provided (N=1 multi-unit in Figure 2B for ArchT under CamK2 promoter, and N=1 penetration in Sup Figure 2d and e for ArchT under CAG promoter for ferrets). Furthermore, these two examples show different response patterns. There is no statistics provided for the size of the effect, and the effects seem inconsistent with each other and with prior literature.

Figure 2B presents recordings from a multi-unit with and without laser (with ArchT delivered under CamK2 promoter). The authors claim that laser presentation reduces neuronal activity. Indeed, there is lower firing rate in middle as compared to top panel during the laser. However when the laser is turned off, there is no increase in neuronal activity in the middle panel (at 1-2 s post tone onset). This is consistent with another sample recording from supplementary figure 3f bottom panel, where there is a prolonged suppression of firing (albeit after a rebound spike which is not observed in Figure 2B), at a higher laser power. However, this result is at odds with the timecourse of effects reported in literature for ArchT which shows recovery <1s post laser onset (for example Han et al, Frontiers, 2011 Figure 2C; Johanssen et al., PNAS, 2014, Figure 1; also under CamK2 promoter).

Furthermore, it is unclear whether the data in the middle panel of 2B are taken from a different time period in the recording than the data in top panel, such that the reduction in firing rate could be due to any number of other effects, such as electrode drift, change in

animal level of arousal, some other external signal. It would be helpful to see a more complete time course starting at 2 s pre tone/laser and ending when the activity returns to baseline (according to Sup Figure 3f, this would be 1-2 sec post laser offset) and demonstrate the time course of the reduction of the response. It would be informative to add results of laser-only (no tone) effects. It would also be reassuring to understand the effects across more units so that statistical analysis can be applied.

In Sup Figure 2, under CAG promoter, there seems to be no reduction in spontaneous firing rate, but in fact reduction of tone-evoked response only. This effect is different from Figure 2B. Oddly, in infra-granular layers, the response seems to increase -- is this statistically significant? Also, the legend does not mention this -- "The reduction in firing ... in general, was less at deeper locations within A1." In fact, there seems to be an increase in responses at deeper locations. Furthermore, the lack of reduction of spontaneous activity seems at odds with the results in Sup Figure 3a, where there is a complete suppression of firing during laser presentation across different intensities -- one would expect a corresponding reduction in spontaneous activity (analogous to the effects in sup 3f vs figure 2 for ArchT under CamK2 promoter).

The main aim of the study was to explore changes in auditory perceptual learning as a consequence of silencing the high-frequency part of the primary auditory cortex during sound-localization training. Therefore, we focused on behavioural analyses, supplemented by electrophysiological and anatomical data to validate our optogenetic design by demonstrating suppression of cortical activity during light delivery and by identifying ArchT-positive cells in the cortex tagged with GFP, respectively.

We agree with the reviewer that it is essential to understand the effects of ArchT on cortical activity, and to that end we carried out both *in vitro* and *in vivo* electrophysiological experiments to address the time course and spatial extent of inactivation. We now provide more information about the example recordings shown in the paper, and have analysed the changes in neural activity over the entire population of neurons recorded in 6 animals, thereby allowing us to include the requested statistical analyses.

We do not think that the effects on neural activity reported by us after illumination of ArchT expressing neurons are unexpected or at odds with previously published reports. For example, Han et al. (2011) reported that delivering light to ArchT-expressing neurons in monkey cortex results in a rapid decrease in a spiking activity, but a much slower recovery after light cessation, which took a median time of 740 ms and sometimes as long as 4260 ms to reach baseline firing rate. Indeed, their Figure 2Bii shows an example neuron with no spiking activity within the first ~500 ms after the light was turned off.

Our *in vitro* recordings, which were carried out at room temperature, allowed us to show unequivocally that ArchT mediates rapid hyperpolarization and near-complete silencing of action potential firing, and that the latency of post-illumination recovery in firing can vary between cells as well as with the magnitude of the light pulse (Supplementary Fig 3a,f). These temporal dynamics are similar to those reported for *in vivo* recordings in monkey cortex by reported by Han et al. (2011).

We show two examples of *in vivo* multi-unit recordings from ferret auditory cortex, one in which AAV8/CaMKII-ArchT-GFP was expressed (Fig. 2B) and the other with AAV8/CAG-ArchT-GFP (Supplementary Figure 2). Both show a clear suppression of driven activity when laser illumination was paired with the auditory stimulus. Again consistent with Han et al. (2011), we did not observe clear evidence *in vivo* for rebound activity on cessation of illumination. Figure 2B shows that both spontaneous and evoked activity are suppressed (the presentations are aligned vertically according to the sound level used, as indicated by the vertical wedge), with baseline firing remaining at a very low level for the next second

following cessation of the light (consistent with the whole-cell recording shown in Supplementary Figure 3f). This is not due to a drift in the position of the electrode or any other change in recording conditions: we interleaved laser-on and laser-off conditions, so we can be certain that the reduction in firing when the laser was on does indeed reflect a suppression of neural activity. Furthermore, we are able to conclude that the firing rate has at least partially returned by 1 second following light cessation, since we used an inter-stimulus interval of 1 second. Any longer-lasting effect of the laser would have suppressed neural activity at the start of the interleaved no-laser sweeps, which was not the case (Figure 2b, raster plot). We apologize that the figure legend did not make this clear and gave the incorrect impression that the laser-on and laser-off data were obtained in separate blocks. We still think it is much easier to see the suppression of firing if the dot raster is arranged in this fashion, but we now provide a fuller explanation in the figure legend and have added a quantification of the change in firing rate between laser-on and laser-off conditions.

Supplementary Figure 2 shows a depth-dependent reduction in the sound-evoked response, but no apparent change in baseline activity. The *in vivo* effects of illuminating the cortex depend on multiple factors, including the size and location of the viral injection, the percentage of cells transfected and also light penetration/intensity. Moreover, Han et al. (2011) showed that the firing rate of a minority of neurons is increased by ArchT activation. It is therefore not surprising that the effects of light delivery on multi-unit activity recorded *in vivo* are not as clear as the *in vitro* whole-cell recordings. Nevertheless, both multi-unit examples in the paper demonstrate the desired suppression of A1 responses, as does the statistical analysis based on the full set of recordings.

The following changes have been made in the manuscript to deal with these points:

Page 4 (paragraph 2):

“Suppression of cortical activity during light delivery was demonstrated electrophysiologically, both *in vivo* in ferrets with each viral construct (Fig. 2b, Supplementary Fig. 2) and *in vitro* in mice (Supplementary Fig. 3). Multiunit responses to auditory stimuli were recorded at 95 locations in the high-frequency region of ferret A1. Acoustically-evoked activity was significantly reduced by laser illumination ($t\text{-test}_{(40)} = 5.791, P < 0.0001$) across all recording sites ($n = 46$) for penetrations within the region where viral injections had been made and where the fiber-optic cannula was within $\sim 800 \mu\text{m}$ of the recording probe. No changes in the firing rate of cortical cells were found at distances greater than this or when the recording probe and/or the fiber-optic cannula were not located within the viral vector injection site (Supplementary Fig. 2). Activity suppression had a rapid onset and often a much slower recovery (Fig. 2b, Supplementary Fig. 3) that resembled the temporal dynamics of ArchT-mediated inactivation described in monkey cortex¹⁸.”

Figure 2 legend:

“**b** Peristimulus time histograms showing multi-unit responses recorded in A1 at the depth indicated by the asterisk in **a** to presentations of 200-ms noise bursts (gray bars). Trials in which the noise bursts were presented alone (top panel, laser off) or paired with the delivery of a 1000-ms light pulse to the implant (green bar) (middle panel, laser on) were interleaved. Spiking activity was reduced (84.6%, $t\text{-test}_{(59)} = 6.19, P < 0.001$) by light delivery. For clarity, the dot rasters in the bottom panel are grouped separately for laser-off and laser-on presentations and show the timing of spikes evoked following presentation of noise bursts of increasing level (indicated by the vertical wedges).”

Methods page 15 (paragraph 4):

“We recorded multi-unit activity using single-shank silicon probes (Neuronexus, Ann Arbor, MI), with 16 recording sites spaced at $150 \mu\text{m}$ intervals in two cases (Figure 2,

Supplementary Fig. 2), and four shanks separated 500 μm with 32 recording sites spaced at 50 μm in four cases (Table S1).”

Methods page 18 (paragraph 3):

“Recordings were made at room temperature using a Multiclamp 700B amplifier (Axon Instruments, CA, USA) with a sampling rate of 10 kHz and digitized by pClamp software (Axon), and later analyzed off-line.”

Methods page 18 (paragraph 3):

“Membrane potential and interspike intervals were measured in 1-s time windows and compared to pre-illumination (baseline) values (ANOVA); the latency for recovery to baseline conditions was established when statistical differences were no longer detected.”

Supplementary Figure 2 legend:

“e Change in multi-unit spiking activity for all recordings made in this animal at each of the 16 recording sites (separated by 150 μm) on the probe, which was positioned perpendicular to the cortical layers. To quantify the effect of laser light delivery, the number of spikes within a window from 15-220 ms (5-ms bin size) following sound onset was compared (*t*-test) between presentations in which the sound was presented alone and those in which it was paired with illumination of A1 for all recording sites in each electrode penetration. Overall, firing was suppressed by 41% across cortical depth for individual penetrations (median, inter-quartile range: 32.9%, 15.2-62.2%, respectively). The largest activity changes were observed superficially, in the vicinity of the implanted cannula (spatial range \sim 800 μm).”

2. The language in the paper that described the behavior is somewhat inconsistent. Adaptation and perceptual learning are used somewhat inter-changingly. I am not sure that either "adaptation" or "perceptual learning" is appropriate here, since the effects are so long-term. Perhaps it's "habituation" (see Kato et al., Neuron 2015). Similarly, the statement that this is a first study demonstrating the "causal role of A1" in learning is somewhat overstated - there are some studies using cooling, as well as pharmacology or optogenetics (e.g. Aizenberg, NN 2013, Aizenberg, PLoS Biology, 2015) as well as the authors' own prior work.

We respectively disagree about the terminology used to describe the changes in sound localization accuracy. Habituation is a form of learning in which the responses to a stimulus decrease or cease after repeated presentations. Consistent with this, Kato et al. (2015) showed that daily passive sound exposure causes a long-lasting reduction in cortical responses to those sounds, which was reversed when the mice subsequently engaged in sound-guided behaviour.

What we are looking at here is a change in the way the brain processes auditory spatial cues, which enables compensation for the reduced hearing in one ear. In other words, the recovery in sound localization accuracy is based on adaptation to the abnormal relationship between those cues and directions in space. This process involves a change in the weighting of different cues (the animals learn to rely more on the intact spectral cues provided by the non-occluded ear) and a compensatory shift in sensitivity to binaural cues. The term “adaptation” for this type of phenomenon has been in use since the initial experiments on monaural occlusion in developing barn owls by Eric Knudsen’s group (e.g. Mogdans & Knudsen (1992) Adaptive adjustment of unit tuning to sound localization cues in response to monaural occlusion in developing owl optic tectum. J Neurosci 12(9):3473-84). The term adaptation is also universally used to describe such effects in studies of human hearing (e.g. Mendonça (2014) A review on auditory space adaptations to altered head-related cues. Front Neurosci 8:219). It is also correct to describe adaptation to altered spatial

cues as a form of perceptual learning since this depends on specific behavioral training, as demonstrated in this and other studies. Again, this is widely accepted in the literature (e.g. Wright & Zhang (2006) A review of learning with normal and altered sound-localization cues in human adults. *Int J Audiol* 45 Suppl 1:S92-8).

In terms of previous work demonstrating a "causal role of A1" in learning, our own previous studies in which either the whole of A1 (Nodal et al., 2010) or specifically layer 5 corticocollicular neurons (Bajo et al., 2010) were lesioned are cited in the Introduction. The Aizenberg et al. studies from the Geffen lab explored the role of inhibitory interneurons in auditory cortex in a differential auditory fear conditioning paradigm, which likely involves different circuits (through the amygdala) from those underlying the positive conditioning paradigm that we have used, but nonetheless demonstrate a role for auditory cortex in a form of learning. Of more relevance is the recent study by Caras & Sanes (2017) showing that pharmacological inactivation of auditory cortex in gerbils reduces learning on an amplitude-modulation detection task without affecting detection thresholds. Citations to this (and other additional) papers have now been added to the Discussion and we are now more specific about the novel contributions made by our current study (pages 9-10):

“Previous studies have shown that training-dependent improvements in sound detection or discrimination are associated with plasticity in the response properties of A1 neurons²⁷⁻³¹, which is likely driven by top-down inputs³¹. Furthermore, direct activation of sensory cortical areas can induce temporary behavioral improvements in the absence of training³²⁻³⁴. Cortical stimulation³⁵ or activation of neuromodulatory inputs³⁶⁻³⁸ can also enhance perceptual learning, while suppression of A1 activity disrupts learning³⁹. Of particular relevance to our results, auditory training has been shown to compensate for deleterious effects of altered experience during infancy on A1 response properties⁴⁰ and behavioral sound detection thresholds⁴¹. Our findings provide further evidence for the role of training in restoring auditory function and for a critical role for A1 in learning. More specifically, they demonstrate that for normal learning to take place, cortical activity is specifically required when auditory stimuli are presented during training and that intact cortical activity between training sessions is not sufficient. It has been argued that transient alterations in the balance of cortical excitation and inhibition contribute to learning and changes in cortical response properties⁴². We have shown that activity in excitatory neurons is required for adaptation to asymmetric hearing loss, but plasticity of inhibitory synapses is also likely to be involved, with these changes triggered by neuromodulatory inputs that likely transmit reinforcement signals to auditory cortex^{43,44}.”

3. For controls, two groups seem to be pulled together -- was there any difference between the groups?

No systematic behavioral differences were found among the control animals, which included animals used to validate the ArchT-dependent specificity of the optogenetic suppression of cortical activity (i.e. light delivery only without transfecting cortical cells with the viral construct and injecting AAV8/CAG-GFP as a control for the viral vector injections), as well as animals that did not receive surgery. Because reviewer #3 had concerns that the control data shown in the original Fig. 6a were based on only 4 animals, we have now run another 5 controls, resulting in a much more robust and consistent dataset.

We have also reorganized Supplementary Table 1 following reviewer #3's suggestion (minor comment 11), separating the animals into experimental and control groups, with details provided about the types of promoters used, the animals used for recordings or anatomy, and the contribution of each animal to specific figures.

4. The results presented in Figure 6 are difficult to interpret. When we compare the performance of Laser Off groups and Laser 2 groups to Control 2 group, what is different? Is final performance or slope different? Furthermore, in laser 1 group, would the same final performance level be reached if the animals were habituated for extended period of time? It also seems that the rate of learning is similar for Control 2, Laser 1, Laser Off and Laser 2 groups -- what is the mechanism that would be consistent with this pattern?

There are no differences between Laser off and Laser on 2, indicating that perturbing cortical activity no longer has any effect on the localization accuracy of the animals (Panel c). By contrast, the performance of these animals was significantly different from that of the Control 2 group. The control animals achieved higher scores at both the start and end of the period of monaural occlusion, so the more appropriate comparison is to compare their overall performance, rather than focus on just their final performance or the slope of the regression line. We have expanded the results section on page 7 (paragraph 2):

“...When compared with the second period of monaural occlusion in the control group, plugging one ear during this second training block produced a greater initial drop in localization performance in the ferrets expressing ArchT, and their overall performance was significantly worse (GLM, $P < 0.00001$, Table 1), with lower scores achieved by the end of the earplug training run (Fig. 6a, b). Thus, optogenetic suppression of cortical activity not only impairs auditory spatial learning, but also results in less effective adaptation when the active auditory cortex is subsequently challenged by monaural occlusion.”

Our interpretation of this is that the initial learning to adapt to altered spatial cues in the control animals (i.e. Control 1) leaves a memory trace that can be retrieved, thereby resulting in a smaller initial deficit at the start of the second earplugging run, with less adaptation then required to achieve the same recovery in performance. That the Laser off and Laser on 2 groups show a similar rate of adaptation (i.e. the learning curves have comparable slopes) as (but perform much worse than) the Control 2 group is consistent with improvements in sound localization then being independent of A1 in each case. What differs between the groups is that the extent of the initial, A1-dependent learning determines the performance level that can be achieved during further training with altered spatial cues. We have expanded the section of the Discussion on page 10:

“...Thus, in the control group, a much smaller initial deficit was induced by occluding the same ear for a second time, with correspondingly less adaptation required to approach pre-plug levels of localization accuracy. This implies that cue reweighting takes place more readily following the initial training with altered localization cues. Similarly, the more limited adaptation observed during optogenetic suppression of high-frequency A1 seems to determine the extent of the plasticity induced by subsequent training with the laser off. This reduced capacity to accommodate abnormal inputs with further training was unaffected when cortical activity was again perturbed by turning the laser on, suggesting that after the initial learning phase, A1 may no longer contribute to experience-dependent plasticity. This could then explain why both control and ArchT animals show a comparable rate of adaptation, with the much higher scores achieved by the controls reflecting the greater learning that occurs during the first earplugging run with normal A1 activity.”

Regarding the question as to whether an extended period of monaural occlusion would result in the Laser 1 group of ArchT animals reaching the same final performance level as the controls, we don't think this would be the case. This is because data from individual animals indicates that, in some cases, their highest scores were achieved before the final

day of earplugging and remained at around that level until the earplug was removed. Unfortunately, we are not able to test this further since our animal protocols do not allow us to maintain the animals on a water-regulated procedure for longer than fourteen consecutive days. However, data from Kacelnik et al. (2006, Figure 4) demonstrate that what matters is the frequency rather than the number of days of auditory training, with ~1,000 trials needed for near-complete adaptation to occur. We replicated this training regime and found clear differences between the control and ArchT animals in the extent and rate of learning.

Minor comments:

1. Page 2 paragraph 3, insert "cues" into "monaural spectral cues" provided by the normal-hearing ear

Added as requested.

2. Title of figure 3 – “Immunolabeling of GFP in a ferret brain transfected with the AAV8/CAG-ArchT-GFP construct. GFP staining (dark black/brown) revealed using primary antibodies against GFP and the DAB-ABC detection kit was always and exclusively visible in cells expressing ArchT”. Not sure whether that is what is shown in the figure. There is no stain of ArchT and GFP to show that GFP was always and exclusively visible in cell expressing ArchT. GFP expression is used to infer ArchT expression.

Absolutely right. The sentences have been modified accordingly:

“Immunolabeling of GFP in a ferret brain (F1115) transfected with the AAV8/CAG-ArchT-GFP construct. GFP staining (dark black/brown) was used as a reporter of ArchT expression.”

3. Page 8, first paragraph: “In keeping with the data shown in Fig. 1...”

This has been changed to “Consistent with the data shown in Fig. 1, ...” (now page 5)

4. Fig 4a, legend can be clarified -- wasn't there laser shined in controls as well? By Laser On, you mean ArchT-Laser on?

The data shown in Fig. 4a came from control and 'ArchT laser on' animals during Pre-Plug sessions. We have amended the figure and figure legend to make this explicit. (As stated in the definition of the control group on page 6, this included animals in which laser light was delivered to the implant without expressing ArchT in the cortex. Not all controls were treated in this way, but we observed no differences in performance between them).

a Sound localization in the session before the right ear was plugged shown by plotting the mean percentage correct scores (and 95% confidence intervals) for 1000-ms bursts of broadband noise at each of 12 loudspeakers positioned at equal intervals in the horizontal plane (0° is directly in front). Data are from ferrets in which left A1 activity was suppressed by illumination of ArchT during each stimulus presentation (green, n= 13) and from a control group (black, n= 13).

Reviewer #3 (Remarks to the Author):

Manuscript ID: NCOMMS-18-13273

Title: Silencing cortical activity during sound localization training prevents auditory perceptual learning

Authors: Victoria M. Bajo, Fernando R. Nodal, Clio Korn, Alexandra O. Constantinescu, Edward O. Mann, Edward S. Boyden, Andrew J. King

Comments to the Authors:

It is well documented that with appropriate training, animals and humans can adapt to unilateral hearing loss, and auditory cortical activity plays an important role in this process. The present study sought to determine whether auditory cortical activity is necessary during training (as opposed to simply throughout the period of unilateral adaptation) to promote sound localization recovery. The authors optogenetically silenced the high frequency region of left primary auditory cortex in adult ferrets, and tested their ability to adapt to a unilateral earplug. While optogenetic silencing had no effect on sound localization accuracy in normal-hearing animals, it impaired adaptation to the earplug. While well-written and clear, there are several aspects of the manuscript that could be strengthened. It is my hope that the comments below will be useful to the authors during this process.

Major:

1. More information should be provided for the behavioral data in Figure 4. Specifically, what kinds of errors did the animals make after earplugging, and how did the error profile change during adaptation? When they made errors, were the animals close (localizing the sounds to an adjacent stimulus location), and were there any differences between the groups? Addressing these questions would allow for a more nuanced interpretation of the data. For example, one possibility is that the poor adaptation in the optogenetic animals is simply explained by the fact that they started out with larger errors. If this were the case, we might expect to see that the error magnitude decreases at the same rate in both groups, but because the control group started with smaller errors, they are able to correct them earlier in training. Alternatively, both groups may display similar error magnitudes at the beginning of training, and the optogenetic silencing may actually impair the rate of error correction. One approach to help distinguish between these possibilities for the reader would be to show representative data for individual animals from each group at different stages of training in the form of stimulus location vs. response location plots (as in Kacelnik et al., 2006, Figure 1A).

We have added a new panel in figure 4 (panel d) depicting the change in error magnitude for the ArchT and control groups before, during and after the first monaural occlusion run. This is based on incorrect trials only to avoid conflating any change in the distribution of the errors with the progressive improvement in localization accuracy (i.e., proportion of correct responses). These data reveal that the error magnitude declines slightly and at the same rate in both groups (comparison of slopes: ANOVA, $F(1, 256) = 0.307$; $P = 0.580$), but that the 'ArchT laser on' animals consistently perform less well, as indicated by the difference in intercept (ANOVA, $F(1, 257) = 173.886$; $P < 0.0001$). In terms of learning, the key difference between the groups is seen in the rate of improvement in the proportion of correct responses shown in Fig. 4b,c. This is also apparent from the stimulus-response plots for individual animals, which we added at the reviewer's request in the new Supplementary Figures 4 and 5.

Figure 4 legend:

“**d** Magnitude of the localization errors on incorrect trials before, during and after this period of monaural occlusion. Symbols in b and d represent data from individual animals, and the lines and shaded areas are the best linear fits and 95% confidence intervals of these fits, respectively, over the 10 days of monaural occlusion.”

Page 5-6:

“**Consistent with** the data shown in Fig. 1, the control group and the animals in which each stimulus presentation was paired with optogenetic silencing of A1 neurons localized the broadband noise bursts equally accurately in the session carried out prior to monaural occlusion (Fig. 4a, b; preplug scores 0.94 ± 0.09 vs 0.90 ± 0.06 , respectively, $t\text{-test}_{(24)} = 1.149$; $P = 0.262$). As soon as an earplug was inserted in the right ear, the proportion of correct responses fell to ~ 0.2 in all animals (**controls 0.26 ± 0.09 ; ArchT 0.17 ± 0.05**) and the error magnitude increased (**controls $57.21 \pm 13.72^\circ$; ArchT $87.44 \pm 12.76^\circ$**), confirming the importance of binaural cues for this task (Fig. 4b-d and **Supplementary Figure 4**).

Daily sound localization training with the earplug in place throughout had a different effect on the localization accuracy of the two groups of animals. Control ferrets adapted with daily training, showing an almost complete recovery in the accuracy with which they localized broadband noise bursts over 10 days of continuous monaural deprivation (Fig. 4b-d black symbols and lines, **Supplementary Figure 4**). **This was manifest as an improvement in localization accuracy (slope significantly different from 0; ANOVA, $F_{(1,128)} = 204.534$, $P < 0.0001$) and a reduction in error magnitude (slope significantly different from 0; ANOVA, $F_{(1,128)} = 10.014$, $P = 0.002$) with training.**

A different result was found during optogenetic suppression of A1. Although they localized normally when intact binaural cues were available, suppression of cortical activity impaired the ability of these animals to adapt to a unilateral earplug (Fig. 4b-d, green symbols and lines, **Supplementary Figure 5; see Table 1 for detailed differences between groups**). **Both localization accuracy (slope significantly different from 0; ANOVA, $F_{(1,128)} = 36.282$, $P < 0.0001$) and precision (error magnitude slope significantly different from 0; ANOVA, $F_{(1,128)} = 15.636$, $P < 0.001$) improved over the 10 days of monaural occlusion in the ArchT animals. However, while the error magnitude decreased at the same rate in both groups (comparison of slopes: ANOVA, $F_{(1,256)} = 0.307$; $P = 0.580$), the localization precision of the ArchT group was worse throughout the period of monaural occlusion (difference in error intercepts: ANOVA, $F_{(1,257)} = 173.886$; $P < 0.0001$) and the rate at which their localization accuracy improved was less than that of the controls (ANOVA, $F_{(4,580)} = 5.863$, $P < 0.0001$).”**

2. The authors use the data presented in Figure 6a to conclude that “auditory spatial learning leaves a memory trace in the brain that facilitates adaptation to a second period of monaural occlusion.” However, the data in Figure 6a appear to come from only 4 animals, and the effect in question looks to be largely driven by data from only one of those 4 animals. They go on to suggest that “optogenetic suppression of cortical activity ... results in less effective adaptation when the active auditory cortex is subsequently challenged by monaural occlusion” (lines 278-280) and they come to this conclusion by comparing the “Control 2” data in Figure 6a with the “Laser Off” data in Figure 6b. A larger sample size for the control group is needed before either of these conclusions can be confirmed.

We agree and have now obtained data from a new group of five controls with two sequential earplugging runs. Data for the new animals have been incorporated in Fig. 4b-d, Fig. 5, Fig. 6a, supplementary Figs. 4 and 6a,c,e, and are described in the accompanying text (with updated statistical analyses). The larger dataset confirms our previous conclusion that control ferrets adapt more readily to a second period of monaural occlusion.

Page 7, first paragraph:

“**Given that adaptation involves a change in the way auditory spatial cues are processed in the brain**, we tested the hypothesis that auditory spatial learning leaves a memory trace in the brain that facilitates adaptation to a second period of monaural

occlusion. This was confirmed by occluding the same ear for a second time in control animals that had previously adapted to the unilateral hearing loss. A much smaller initial deficit was observed when the ear was replugged (proportion of correct responses 0.47 ± 0.18) than when the animals first experienced an earplug (0.26 ± 0.09) (Fig. 6a). Furthermore, most of the control ferrets achieved their maximum score by ~day 5 and remained at around that level until the end of the second period of monaural occlusion. Consequently, they achieved higher scores (Table 1) and the slope of the adaptation function was flatter (ANOVA $F_{(4,580)} = 5.863$, $P < 0.0001$, post hoc, $P=0.013$) than when they were first trained with one ear occluded.”

3. For the sake of scientific transparency and replicability, more methodological details are needed about the virus injections. For example, what was the flow rate used for the injections? What equipment was used to perform the injections? How long were the pipettes or needles left in place after each injection before removing them? Similarly, important details are missing about the behavioral experiments. How many trials were presented in each session of the earplugging experiments? Did optogenetic and control groups experience the same number of trials over the same number of sessions? How often were the sessions? (Once every 24 hr? 48 hr?) These details are critical to convince the reader that the experimental and control groups received an equivalent degree of training, with similar opportunities for memory consolidation between training sessions.

This information has been added to the Methods:

Pages 11-12:

“The injections were performed using a pressure injector (Nanoejector, Drummond Instruments). The volume of each injection was 50.6 nL, made using glass micropipettes with a 15-20 μm tip (F1217 and F1411, and the 18 first cases in Supplementary Table S1). The pipette was left in place at least for 5 minutes followed each injection. The post-injection survival time was up to 30 months in the ferrets used for behavior, and 4 weeks in the animals used for recordings and/or anatomy (F1115, F1431, F1003, F1110, and F1120; Supplementary Table S1). In all cases, injection sites and sizes were very similar, with comparable GFP-positive terminal fields in the MGB and IC.

AAV8/CAG-ArchT-GFP and AAV8/CaMKII-ArchT-GFP were also injected in the motor cortex in five mice. Two of these animals were used for *in vitro* patch-clamping experiments 4 weeks later, one with each promoter (Supplementary Figure 3), two were used as anatomical controls, and the last one used to confirm virus expression 8 months later. In each case, injections were made at different cortical depths in a single penetration.”

Page 13, paragraph 2:

“Experimental (ArchT) and control groups performed a similar number of sessions and trials. The ferrets were trained in blocks of five days for normal sound localization measurements and in blocks of fourteen days for the earplugging experiments. During these periods, they were tested twice a day and the number of trials animals performed ranged from 50-100 trials per session. Each animal performed a minimum of 300 trials per sound duration under normal hearing conditions and a minimum of 150 trials per day when wearing an earplug. In a small percentage of cases (~15%) and especially in the first few days of monaural occlusion, an additional session was sometimes required to reach the minimum number of daily trials.”

Minor:

1. The title of the manuscript is a bit overstated. Silencing cortical activity does not prevent perceptual learning. Rather, silencing impairs or impedes learning. I suggest changing the title to more accurately reflect the reported findings.

Agreed. “Prevents” has been substituted by “impairs” in the title.

2. I believe there is a missing word on line 57: “..increased dependence on the monaural spectral cues provided by the normal-hearing ear...”

Thank you. The text now states “... **increased dependence**”. Page 2, line before last in the revised version.

3. “In particular, it is not known whether behavioral plasticity is driven by exposure to altered inputs throughout the period of sensory deprivation or specifically by the activity patterns induced by training.” This statement (lines 65-67) is not quite true, and is self-contradictory. As the authors themselves point out, adaptation depends on training, suggesting that mere exposure to altered inputs is insufficient to drive behavioral plasticity. I suggest revising this statement, focusing more on the question of when auditory cortical activity is necessary for behavioral adaptation.

This sentence referred specifically to when cortical activity is required, rather whether training or passive exposure to altered cues is needed for behavioral adaptation to take place. Our previous work with permanent lesions could not distinguish these possibilities. We have changed the sentence to make this clearer (Page 3, paragraph 1):

“In particular, it is not known whether cortical activity following sound presentation during training is sufficient for adaptation to take place or whether A1 is also required for learning retrieval when abnormal spatial cues are experienced again.”

4. As seen in Figure 2b, and Supplementary Figure 3a and 3f, the effects of laser stimulation can persist for up to several seconds, raising the possibility that cortical activity generally does not return to baseline before the next trial starts. What was the intertrial-interval during behavioral testing? Was it longer than the average length of time that the light-evoked effects persisted?

This is an important point. One of the strengths of our behavioral setup is that the animal controls the task progression and the only time restriction applied is a 500-1000-ms variable waiting time at the central platform prior to stimulus presentation. However, because this is an approach-to-target task (70 cm radius), and a response is required before the next trial can be initiated, the minimum time between stimulus presentations is approx. 4 - 4.5 s. This is a conservative value as it is based on correct responses (incorrect response times are longer; page 6; see Nodal et al., 2008 for a full description of response times in this task). This inter-trial interval range is much longer than the recovery time for cortical activity indicated in Figure 2b and Supplementary Figure 3a and 3f and in previous studies with ArchT (e.g. Han et al., 2011). We are therefore confident that normal neural activity would have returned by the start of the next trial. We have included the following statement to make this explicit (end of page 4, top of page 5):

“The inter-trial interval was long enough ($\geq 4-4.5$ s) to ensure that any long lasting light-evoked change in neural activity (Fig. 2b) was over before the next stimulus presentation.”

5. The opening sentence of the legend for Figure 3 is confusing. The figure itself does not attempt to demonstrate using immunohistochemistry that GFP and ArchT are co-localized. Co-localization is assumed from the viral construct. However, the opening sentence states that “Immunolabeling of GFP... was always and exclusively visible in cells expressing ArchT.” I suggest revising this sentence.

The reviewer correctly makes the same point as reviewer #2. The sentence has been changed and now reads:

“Immunolabeling of GFP in a ferret brain (F1115) transfected with the AAV8/CAG-ArchT-GFP construct. GFP staining (dark black/brown) was used as a reporter of ArchT expression.”

6. More information should be provided for the stereological estimations reported in the legend of Figure 3. Do the numbers represent data pooled across animals used for behavioral testing and anatomy? If so, are there differences in the transfection estimates at 4 weeks post-injection, compared to 30 weeks post injection? How many animals contributed to each of these estimates?

Detailed information about the stereological estimations has been added to the legend of Figure 3, the Methods, and in the new Supplementary Table S1. Data from animals used for behavior, anatomy and recordings were pooled together because no difference in the transfection estimates was found at 4 weeks compared to 30 weeks post injection.

Stereological estimations were made in five animals where GFP and NeuN immunostaining were permanent using the ABC system and performed on consecutive sections (one for GFP and one for NeuN). Of those animals, three were injected with the construct with the CAG promoter and two with the CaMKII promoter. In one additional case (F1217, see Supplementary Table S1), sections were also immunoreacted to visualize GFP and NeuN positive cells with ABC, but the transfection did not work and no positive GFP cells were found. The sound localization behaviour of this animal was identical to that of the control cases and it was therefore treated as a behavioural control in our study, demonstrating that in the first round of monaural occlusion, light delivery does not have any effect if the cortical cells are not expressing ArchT.

In another fourteen ferrets (ten used for behavior, one for electrophysiological recordings, two for anatomy only, and one as a behavioral control injected without ArchT), which are indicated in the Anatomy column in Supplementary Table S1 as FI (fluorescence), GFP-positive cells were identified using fluorescence and confocal microscopy. In these cases, no stereological estimates were performed, but we always checked using confocal microscopy that injection sites had the same location and similar sizes as the cases reacted with ABC, and that GFP-positive axons and terminal fields were found in the contralateral auditory cortex, ipsilateral medial geniculate body and in the inferior colliculus mainly ipsilaterally (as illustrated by the example in Fig. 2).

Text added to Figure 3 legend:

“Stereological estimations made using the optical fractionator probe (n = 5 cases, 3 using CAG and 2 using CaMKII promoters, optical dissector: 1/8th of the sections, grid size 0.09 mm² and a 100 x 100 μ m counting frame. Coefficient of error representing the precision of

the estimations <0.05), revealed that when AAV8/CAG-ArchT-GFP was used, $84.4 \pm 14.8\%$ of the neurons in the injection site were transfected, whereas, with AAV8/CaMKII-ArchT-GFP, the transfection rate was $60.03 \pm 14.8\%$.”

Text added to Methods (pages 16-17):

“Out of 20 ferret brains (13 ArchT cases with behavior, 2 with recordings only, 3 with anatomy, plus 2 controls), fourteen were used to visualize GFP fluorescence on the sections and six were used for permanent immunoreaction of GFP and NeuN positive cells using the DAB-ABC system (F1 and ABC, respectively; anatomy column in Supplementary Table S1). In the first group (F1), one set of sections was Nissl stained with 0.5% cresyl violet to identify the limits of different structures in the ferret brain, a second set was mounted on gelatinized slides, dried, and coverslipped for examination under a fluorescence microscope, and the last two set of sections were incubated with rabbit GFP primary antibody followed by incubation in Alexa Fluor secondary antibody (anti-rabbit Alexa Fluor488 for GFP detection, Molecular Probes dilution 1:200; Fig. 2a, c-e). Using confocal microscopy (confocal LSM 710 Carl Zeiss Microimaging microscope), we verified that the location and size of the injections sites in the auditory cortex and the anterograde labeling of axons and terminal fields in the auditory thalamus and midbrain injections were comparable in all animals.

The second group (ABC) was used for stereological analysis of GFP-positive cells in the injection site in A1. The first and second sets of sections were respectively stained for Nissl substance and mounted on gelatinized slides, dried, and coverslipped. The other two sets of sections were immunoreacted for permanent staining using the avidin-biotin-complex-system (ABC) with either mouse anti-neural nuclei protein (NeuN) monoclonal antibody to label every neuron in the injection site or an anti-GFP mouse monoclonal antibody to label the neurons transfected with the viral construct and therefore expressing GFP. For these immunoreactions, sections were washed several times in 10 mM phosphate-buffered saline (PBS) with 0.4% Triton X₁₀₀ (PBS-Tx) to permeabilize the cell membranes and incubated in 5% normal horse serum in PB for 1 hour to block unspecific staining. The sections were then incubated for 72 hours at 5°C in the primary antibody (mouse monoclonal anti-GFP, dilution 1:400, Sigma-Aldrich or mouse monoclonal anti-NeuN, Map377, Millipore Corp.), rinsed several times in PBS, incubated with biotinylated horse anti-mouse IgG (H + L) (dilution 1:200, Vector Laboratories), rinsed again several times in PBS, incubated for 90 minutes in avidin-biotin peroxidase (Vectastain Elite ABC kit, Vector Labs), rinsed a final time in PBS, and then incubated with 0.4 mM 3,3'-diaminobenzidine (DAB) (Sigma-Aldrich) and 9.14 mM H₂O₂ in 0.1 M PB until the reaction product was visualized (Fig. 3).

StereoInvestigator Software (MBF Bioscience, MicroBrightField Inc., Williston, VT) was used for stereological estimations of NeuN and GFP neurons. The optical fractionator was chosen as the stereological probe with parameters set to obtain a coefficient of error, which represents the precision of the population size estimate, of <0.05 (optical disector: 1/8th of the sections, grid size: 0.09 mm², counting frame: 100 x 100 μm) (Fig. 3c, d). From an initial group of six cases for stereology, we had to exclude one animal because no GFP expression was present (case F1217, see supplementary Table S1).”

7. “...[T]he learning deficit was not due to a differential effect on excitatory versus non-excitatory neurons...” This statement (lines 234-235) is not necessarily true. In order to determine whether there is or is not a differential effect of excitatory vs. inhibitory cells, one would need to use the CamKII promoter to target excitatory neurons (as the authors do here), use a different promoter to target inhibitory neurons (such as mDlx) and directly compare the effects. As the CAG promoter targets both excitatory and inhibitory neurons, and the effects were similar to the CamKII promoter, the most one can conclude is that

excitatory neurons are involved. One cannot rule out the possibility that inhibitory neurons also play a role, as the authors themselves point out in the discussion (lines 371-372). I suggest revising the statement in question.

We agree and have changed the sentence accordingly (page 6, paragraph 2):

“...Impaired adaptation during optogenetic suppression of A1 was observed with both CAG and CaMKII promoters, indicating that the activity of excitatory cortical pyramidal neurons is essential for learning to occur, without ruling out the possibility that the inhibitory neurons also play a role (Supplementary Figure 5; adaptation slopes for CAG and CaMKII were not statistically different: ANOVA, $F_{(126,1)}=1.354$, $P = 0.247$).”

8. Table 1 does not provide information regarding the contribution of within-session improvement and/or between session consolidation to learning, yet is referenced as such in the text (line 241).

Table 1 only includes the P values and odds ratios of the comparisons between groups in our statistical model. However, the model included one term related to the session number and another related to the order (log transformed to ensure linearity) of each trial within each session to explore how the probability of a correct response during the period of monaural occlusion depended on inter- and intra-session order. Both terms were statistically significant as mentioned in the text, but for completeness we have included the odds ratios for both parameters in addition to the P values. In addition, we have included a new table (Table 2) where odds ratios for the factors of day of monaural occlusion and order of the trial within each session are shown, which does show more directly the contribution of within-session and between-session improvements. The reference to Table 1 has been deleted and the reference for the new Table 2 has been added instead.

9. “[P]erturbing A1 activity whenever sounds are presented as part of a localization training task impairs adaptation...” (lines 304-305). “[W]ithin-trial activity in A1 is critical for learning...” (line 354). These statements do not address the fact that the effect of the optogenetic manipulation can persist well after sound offset, likely into the intertrial-interval. This caveat should be addressed in each of these instances, or the statements should be revised accordingly.

As previously stated in response to minor point 4, the inter-trial interval is $\geq 4-4.5$ s, so we don't think that optogenetic suppression will persist beyond the time that the animal has made its response. However, we agree that this may outlast stimulus duration. We have amended both sentences:

page 8, paragraph 1:

“Our results show that transient perturbation of A1 activity initiated at the onset of the sound presentation during a localization training task impairs the ability of adult ferrets to adapt to altered spatial cues induced by reversible occlusion of one ear.”

page 9, paragraph 2:

“ By taking advantage of the transient nature of optogenetic suppression with ArchT, we were able to show that activity in A1 during the sound-localization task is critical for learning, further highlighting the importance of behavioral training in promoting the recovery of performance accuracy in the presence of hearing loss in one ear.”

10. The discussion of the causal relationship between cortical activity and perceptual learning, and the role of training in restoring sensory function (lines 357-363) should be expanded. Both gain-of-function and loss-of-function studies have investigated the role of various brain regions (including auditory cortex) in perceptual learning, and in sensory restoration. See:

Gain of function and perceptual learning:

Glennon et al. (2018) Locus coeruleus activation accelerates perceptual learning. *Brain Res.* doi: 10.1016/j.brainres.2018.05.048.

Karim et al. (2006) Facilitating effect of 15-Hz repetitive transcranial magnetic stimulation on tactile perceptual learning. *J Cogn Neurosci* 18:1577-1585.

Pleger et al. (2006) Repetitive transcranial magnetic stimulation-induced changes in sensorimotor coupling parallel improvements of somatosensation in humans. *J Neurosci* 26:1945-1952.

Reed et al. (2011) Cortical map plasticity improves learning but is not necessary for improved performance. *Neuron* 70:121-131.

Shibata et al. (2011) Perceptual learning incepted by decoded fMRI neurofeedback without stimulus presentation. *Science* 334:1413-1415.

Tegenthoff et al. (2005) Improvement of tactile discrimination performance and enlargement of cortical somatosensory maps after 5 Hz rTMS. *PLoS Biol* 3:e362.

Loss of function and perceptual learning:

Caras and Sanes (2017) Top-down modulation of sensory cortex gates perceptual learning. *PNAS* 114:9972-9977.

Sensory restoration:

Kang R, Sarro EC, Sanes DH (2014) Auditory training during development mitigates a hearing loss-induced perceptual deficit. *Front Syst Neurosci* 8:49.

Voss et al. (2016) Pairing Cholinergic Enhancement with Perceptual Training Promotes Recovery of Age-Related Changes in Rat Primary Auditory Cortex. *Neural Plast* 2016:1801979.

As requested, we have expanded the discussion about the causal relationship between cortical activity and perceptual learning, and the role of training in restoring sensory function by including the papers mentioned by the reviewer. This also addresses the second major point of reviewer #2:

Pages 9-10:

“Previous studies have shown that training-dependent improvements in sound detection or discrimination are associated with plasticity in the response properties of A1 neurons²⁷⁻³¹, which is likely driven by top-down inputs³¹. Furthermore, direct activation of sensory cortical areas can induce temporary behavioral improvements in the absence of training³²⁻³⁴. Cortical stimulation³⁵ or activation of neuromodulatory inputs³⁶⁻³⁸ can also enhance perceptual learning, while suppression of A1 activity disrupts learning³⁹. Of particular relevance to our results, auditory training has been shown to compensate for deleterious effects of altered experience during infancy on A1 response properties⁴⁰ and

behavioral sound detection thresholds⁴¹. Our findings provide further evidence for the role of training in restoring auditory function and for a critical role for A1 in learning. More specifically, they demonstrate that for normal learning to take place, cortical activity is specifically required when auditory stimuli are presented during training and that intact cortical activity between training sessions is not sufficient. It has been argued that transient alterations in the balance of cortical excitation and inhibition contribute to learning and changes in cortical response properties⁴². We have shown that activity in excitatory neurons is required for adaptation to asymmetric hearing loss, but plasticity of inhibitory synapses is also likely to be involved, with these changes triggered by neuromodulatory inputs that likely transmit reinforcement signals to auditory cortex^{43,44}.”

11. While I appreciate the authors' attempt at transparency, it is often unclear which (and how many) animals were used for which experiments. For example, How many animals contributed to the data in Figure 1c-e? Did the animals that contributed data to Figure 1c-e also contribute data to Figure 4-6? I suggest adding some information to Supplementary Table 1 to make clear the specific figure panels to which each animal contributed. It would also be helpful to organize the table by experimental group, rather than by the animal identity. Finally, I suggest including sample sizes in every figure, either in the panels themselves, or in the figure legends.

Thirteen ArchT animals contributed to the data shown in Figure 1c-e (the first thirteen cases listed in Supplementary Table S1). These animals also provided the data shown in Figures 4-6 (except for Fig 5 and Fig. 6a, where there are 12 ArchT animals, since F1218 only contributed to the first two periods of monaural occlusion: Laser on 1 and laser off) and Supplementary Figures 5-6.

The number of control animals has been increased (also to 13 in total; the last 13 cases listed in Supplementary Table S1). These 13 animals provided the data for the first period of monaural occlusion (Figs 4-6a, Supplementary Figs. 4-6a,c) and nine of them to the second period of monaural occlusion (Figs. 5-6a, Supplementary Fig. 6a,e).

Following the reviewer's suggestion, we have reorganized Supplementary Table 1 by experimental group. Different colour cells identify the different procedures in each case: Behaviour (sound localization – SL, 1-3 periods of monaural occlusion - MO), recordings (*in vivo*, 1 shank or 4 shank extracellular recordings in anesthetized ferrets and *in vitro*, patch-clamp recordings in mouse slices), and anatomy (fluorescence – FI and using the avidin biotin complex – ABC). In addition, an extra column has been added to indicate the specific figure panels to which each animal contributed.

As requested, sample sizes have also been included in every figure legend.

Reviewers' Comments:

Reviewer #1:

Remarks to the Author:

I am very happy with the detailed responses to my queries supplied by the authors, and by their changes to the Results and Discussion sections that have clarified the responses to my concerns.

Reviewer #2:

Remarks to the Author:

The authors did a nice job addressing our concerns, and the manuscript is now much improved. I have no further concerns.

Reviewer #3:

Remarks to the Author:

The authors have done a fantastic job addressing my and my fellow reviewers' comments. Their considerable efforts have resulted in a strong, convincing, and clear manuscript. Well done all around. My remaining comments will hopefully only improve this excellent manuscript even further.

Major:

1. While it is now clear that the trials in Figure 2b were interleaved, I disagree with the authors that the dot raster organization is sufficient for demonstrating optogenetic suppression of neural activity. Providing a full time course for one representative unit, showing pre-sound baseline activity, a sound evoked onset response, and a complete return to baseline, both with and without laser stimulation (as I and reviewer 2 suggested originally) would be the clearest way to demonstrate to your readership that ArchT worked in vivo as expected.
2. The error magnitude on the first day of the initial earplug session seems to differ between control and ArchT animals (57 vs. 87; line 144). Was this difference significant? If so, does this mean that the ArchT animals are more susceptible to large localization errors immediately after unilateral hearing loss? Or do the authors think that some adaptation occurs during the first training session, during which the Controls begin to pull ahead of the ArchT animals? Have they compared within-session performance specifically during the first day of earplugging to see if the latter is the case? Also- are these error magnitudes absolute errors or relative to the initial pre-plug errors? In which figure are these values displayed? In Figure 4d, error magnitudes on the first day of plugging appear to be 90 and 130.
3. It's unclear how the regression slopes throughout the manuscript were statistically compared. For example, the data in Figure 4 (and the methods section) seem to indicate that a single regression was fit to the mean values in each group. Is this correct? If so, how could the slopes of these two fits be compared with an ANOVA, as suggested on line 162 (and elsewhere throughout the manuscript)? Or were the slopes of these two lines actually compared by looking for a significant interaction of training day and group in an ANCOVA? Please clarify.
4. In Figure 6a, the data look bimodal- many animals appear to adapt quickly, in line with the author's hypothesis, but others (at least 3) seem to do poorly, similar to session 1. Could the authors comment

on this? How did these three animals perform in the first session?

5. The authors state that multiple training sessions were run each day. Were data from each session collapsed into a single day for display and analysis? Did any consolidation or improvement occur between training sessions within a given day?

Minor:

1. Line 153: Typo. I'm assuming the authors meant "P= 0.02 or P = 0.002"?

2. Line 355: Typo. "followed" should be "following".

3. Line 490-1: "Laser stimulation was...paired with... sound presentations, beginning either at the same time." Was there another option? Or were all laser stimulations presented at sound onset?

4. Table 1 and Table 2: The legends indicate that significant values are shown in red text, but there is no red text in either table.

5. While the physiology in Supp. Fig 3 is quite convincing, the merged images are not, and do little for making the case that the recorded neuron actually expressed ArchT. It's clearly the authors decision whether to keep the images in the manuscript or not, but in my opinion, they should either be removed, or replaced with better, more convincing images.

Reviewer comments in blue, responses in black, new text in red

Reviewer #3 (Remarks to the Author):

The authors have done a fantastic job addressing my and my fellow reviewers' comments. Their considerable efforts have resulted in a strong, convincing, and clear manuscript. Well done all around. My remaining comments will hopefully only improve this excellent manuscript even further.

Thank you.

Major:

1. While it is now clear that the trials in Figure 2b were interleaved, I disagree with the authors that the dot raster organization is sufficient for demonstrating optogenetic suppression of neural activity. Providing a full time course for one representative unit, showing pre-sound baseline activity, a sound evoked onset response, and a complete return to baseline, both with and without laser stimulation (as I and reviewer 2 suggested originally) would be the clearest way to demonstrate to your readership that ArchT worked in vivo as expected.

We agree that the reviewer's proposed figure does represent the best way to illustrate the effects (magnitude and time-course) of optogenetic suppression on cortical activity. The reason why we did not include such a plot previously is that the protocol we used involved making interleaved presentations of sound alone and sound-plus-laser, with both starting at the beginning of each sweep. This was done in order (1) to make a direct comparison of activity during presentations with and without laser stimulation, and (2) to obtain the maximum number of units while recording from multiple locations in extremely precious animals that had been used for many months of behavioral testing. In pilot experiments, we had previously verified that intervals of 2 seconds between each presentation were long enough for a complete return to baseline activity after laser activation, but agree that not having a period of true spontaneous activity before the laser was turned on leaves uncertainty over exactly how optogenetic suppression affects activity in the ferret auditory cortex.

To address this point properly, we have therefore carried out a new electrophysiological recording experiment (after expressing ArchT in the auditory cortex by AAV injections) in one of the animals previously used as a behavioural control (F1804); this animal therefore appears as both a control and experimental subject in the table of animals used in this study.

In this new recording experiment, we employed longer sweeps and delayed the presentation of the sound stimulus and laser activation until after the start of the sweep, enabling us to observe the full time course of the inactivation as suggested by the reviewer. In addition, we recorded neural activity in separate blocks with the laser either off or on to more closely match the paradigm used during the earplugging behavioral tests. Each sweep therefore had a duration of 5 seconds, starting with 750 ms of spontaneous activity recording, to correspond to the average duration of the variable 500-1,000ms delay after the animal triggered the central spout but before sound onset in each behavioral trial. We then presented the auditory stimulus (1,000 ms broadband noise), either with (laser on) or without (laser off) laser illumination, followed by a further recording period of 3,250 ms, which corresponds to the typical time taken by ferrets to make a response and to return to the central platform for the next trial.

To provide further information about the effectiveness of optogenetic suppression of cortical activity, we included both constructs (AAV8/CAG-ArchT-GFP and AAV8/CaMKII-ArchT-GFP) in this animal by injecting each at different locations of A1 (as illustrated in the new Supplementary Figure 2).

We have included these new electrophysiological data in Figure 2 (new panel e-h, replacing the previous panel b) to illustrate the time course of optogenetic suppression for one neuron (AAV8-CaMKII-ArchT-GFP) in the form of PSTHs (with the laser off or on, panels e and f), raster plots to compare activity in each condition over multiple trials (panel g), plus the raw signal during one of the sweeps with the laser on (panel h). Together, these plots show a clear suppression of activity (to well below the spontaneous level) when the laser was on, including a reduced sound-evoked offset response, and a recovery of spontaneous activity very soon after the end of laser illumination. We believe that these data show as convincingly as published figures for any other species (including monkeys and mice) that optogenetic suppression with ArchT is effective in ferret auditory cortex.

In the new Supplementary Figure 2, we show five examples at different depths from the same electrode penetration in an area of the auditory cortex where AAV8/CAG-ArchT-GFP was injected. Laser illumination was effective in reducing neural activity at all cortical depths, again with a recovery to baseline typically within 1 second of the laser being turned off (and therefore well before the next trial in the behavioral tests).

Quantification of these effects has been added to the text (new text in red)

Page 4, paragraph 2:

“Suppression of cortical activity during light delivery was demonstrated electrophysiologically, both *in vivo* in ferrets with each viral construct (Figure 2e-h, Supplementary Figure 2) and *in vitro* in mice (Supplementary Figure 3). Multiunit responses to auditory stimuli were recorded at 99 locations in the high-frequency region of ferret A1. Acoustically-evoked activity was significantly reduced for units recorded within the region where viral injections had been made and where the fiber-optic cannula was within ~800 μm of the recording probe. No changes in the firing rate of cortical cells were found at distances greater than this or when the recording probe and/or the fiber-optic cannula were not located within the viral vector injection site (Supplementary Figure 2). Activity suppression had a rapid onset and a slower recovery *in vitro* (up to 2 s after turning off the laser, Supplementary Figure 3) than *in vivo* (Figure 2 and Supplementary Figure 2), where baseline activity was typically restored within 0.5 s, resembling the temporal dynamics of ArchT-mediated inactivation described in monkey cortex¹⁸.”

“**Figure 2.** Optogenetic suppression of ferret A1. **a** ArchT-GFP expression in a coronal view of the auditory cortex and thalamus of a ferret brain (F1431) transfected with the AAV8-CaMKII-ArchT-GFP construct. IS, injection site. **b, c** Higher magnification views of the corresponding boxed regions in **a**. **d** GFP-labeled axons and terminals in the inferior colliculus originating from transfected neurons at the injection site shown in **a**. **e-h** Neural activity recorded in a different animal (F1804) from a representative A1 unit expressing ArchT under the CaMKII promoter in response to 1000-ms bursts of broadband noise (BBN, gray bar) presented either alone (**e**, 20-ms binned peristimulus time histogram with the laser off) or paired with a 1000-ms light pulse (**f**, laser on, green bar). **g** The corresponding dot rasters for this unit grouped into laser-off (top half) and laser-on presentations (bottom half). **h** Activity recorded during a single presentation (#55 of the raster plot) with BBN paired with laser illumination (green) is shown in the bottom panel. Activation of ArchT by laser

illumination resulted in a suppression of the unit's sound-evoked response ($t_4 = 6.584$, $P = 0.001$), including a loss of its offset response. Comparable spike rate to that found before BBN/laser illumination was observable as early as 0.5 s after laser offset ($t_4 = 2.271$, $P = 0.086$). Auditory cortical cells continued expressing ArchT 30 months after the viral construct injections. Behavioural testing took place over a 30 month period and was stable throughout, with no impact of when the animals were tested on whether they adapted or not to monaural occlusion. Furthermore, once the brains had been processed histologically, subsequent neuronal quantification showed no differences in the overall number or density of transfected cells with survival time following viral vector injections. **Scale bars in a and d, 1 mm.** AC, auditory cortex; D, dorsal; M, medial; MGB, medial geniculate body.”

“**Supplementary Figure 2. Optogenetic suppression of sound-driven activity in A1. a** Histological section of a flattened left auditory cortex (ferret F1804) showing GFP immunofluorescence associated with ArchT expression using two different promoters. The inset shows a photograph of the cortical surface of this animal on which the locations of the recording sites and optical fiber placements in the dorsal high-frequency region of A1 have been marked. **b** Examples of unit responses recorded at location #7 (AAV8/CAG-ArchT-GFP injection) at 5 different cortical depths, which are indicated by the solid black circles on the schematic of the recording probe (distance between adjacent recording sites, 50 μ m). For each unit, dot rasters show the responses to 100 presentations of broadband noise (BBN, 1000-ms duration), grouped into laser-off (top half, gray) and laser-on presentations (bottom half, green) in the same fashion as during behavioral testing. Each sweep lasted 5 s (comprising 750 ms of spontaneous activity preceding the 1,000-ms stimuli followed by 3,250 ms after laser offset) so that the time-course of optogenetic suppression could be measured. The next two columns show the corresponding peristimulus time histograms (20 ms bins) for BBN only (left) and BBN plus laser illumination (right). All 5 units exhibited a significant suppression of acoustically-driven activity (comparison of activity during a time window from 760-1,760 ms for BBN laser-off vs BBN laser-on presentation, t -test, $P < 0.05$ for each unit). During sound presentation, the firing rate of these example units was reduced by laser illumination to different extents (85%-25%), but in each case the prominent offset response observed in the absence of laser illumination was eliminated when the laser was turned on (comparison of pre-stimulus spontaneous activity vs offset response during a time window from 1,800-2,000 ms; laser-off, t -test, $P < 0.05$; laser-on, $P > 0.165$). The activity of all the units recovered soon after the laser was turned off (comparison of pre-stimulus spontaneous activity vs activity 0.5 s after the laser was turned off, $P > 0.502$ for each unit). **c** Histological sections at the level of the ipsilateral medial geniculate body and inferior colliculus showing the terminal fields of the axons of descending cortical projection neurons expressing GFP. **Scale bars in a and c, 1 mm.** D, dorsal; M, medial; P, posterior.”

2. The error magnitude on the first day of the initial earplug session seems to differ between control and ArchT animals (57 vs. 87; line 144). Was this difference significant? If so, does this mean that the ArchT animals are more susceptible to large localization errors immediately after unilateral hearing loss? Or do the authors think that some adaptation occurs during the first training session, during which the Controls begin to pull ahead of the ArchT animals? Have they compared within-session performance specifically during the first day of earplugging to see if the latter is the case? Also- are these error magnitudes absolute errors or relative to the initial pre-plug errors? In which figure are these values displayed? In Figure 4d, error magnitudes on the first day of plugging appear to be 90 and 130.

All the errors reported are absolute errors. The apparent discrepancy between the error sizes between the text and Figure 4d is because the former refers to the error magnitude for all trials (controls $57.21 \pm 13.72^\circ$; ArchT $87.44 \pm 12.76^\circ$), while the figure shows the error magnitude for the incorrect trials only (controls $89.52 \pm 21.13^\circ$; ArchT $130.77 \pm 18.12^\circ$). Both between-group differences are statistically significant. Plotting the error magnitude for incorrect trials only during the period of monaural occlusion is the right thing to do because this avoids the data being skewed by the progressive increase in the proportion of correct responses (i.e., errors of 0°). Including both measures, however, is obviously confusing, so we have removed any reference to errors in the paragraph on page 5.

To be consistent, we have also changed Supplementary Figure 6 so that this plots error magnitudes on incorrect trials (instead of for all trials)

We did highlight the difference in the error magnitudes between the control and ArchT animals in the previous manuscript version by stating that the intercepts for the linear fits to the error magnitude across the period of monaural occlusion differed between the groups. We have rewritten this section to make this more explicit:

Pages 5-6:

“A different result was found during optogenetic suppression of A1. Although they localized normally when intact binaural cues were available (Figure 4a), suppression of cortical activity impaired the ability of these animals to adapt to a unilateral earplug (Fig. 4b-d, green symbols and lines, Supplementary Figure 5; see Table 1 for detailed differences between groups). **The proportion of correct responses increased** (slope significantly different from 0; ANOVA, $F_{(1, 128)} = 36.282$, $P < 0.0001$) (Figure 4b, c) and the error magnitude **on incorrect trials declined** (slope significantly different from 0; ANOVA, $F_{(1,128)} = 15.636$, $P < 0.001$) (Figure 4d) over the 10 days of monaural occlusion in the ArchT animals. However, while the errors decreased **in size** at the same rate in both groups (comparison of slopes: ANOVA, $F_{(1, 256)} = 0.307$; $P = 0.580$), they **were significantly larger** throughout the period of monaural occlusion in the ArchT group (difference in error intercepts: ANOVA, $F_{(1, 257)} = 173.886$; $P < 0.0001$) (Figure 4d) and the localization accuracy of these animals recovered at a slower rate than the controls (ANOVA, $F_{(4, 580)} = 5.863$, $P < 0.0001$) (Figure 4b, c).

With regard to the interpretation of these data, the larger localization errors exhibited by the ArchT animals on the first and each subsequent day of monaural occlusion might potentially reflect an impairment in spatial processing. But these animals clearly fail to learn when presented with altered cues as well as the controls, as shown by Figure 4b,c, so we agree with the reviewer that the controls start pulling away from day one and continue to do so throughout the adaptation period (compare Supplementary Figures 4 and 5). Indeed, Van Wanrooij & Van Opstal AJ (2007) (reference 7 in our paper) have argued that reweighting of spatial cues occurs as soon as human listeners experience an earplug in one ear. Since up-weighting of spectral cues available at the non-occluded ear provides the principal basis for training-dependent adaptation, this could explain why our control group make smaller errors from the first day of plugging (Fig. 4d). We have added this to the Discussion:

Pages 8-9:

“For the broadband stimuli used in the ear-plugging experiments, adaptation to monaural occlusion results principally from learning to up-weight the spectral localization cues available at the non-occluded ear^{6,8,13}. Ferrets raised with one ear occluded develop greater sensitivity to high-frequency spectral features that contain most of the directional information in the head-related transfer function^{10,13}. Because adaptation in adult animals was greatly reduced by optogenetic suppression of activity in the region of A1 representing those frequencies, it is likely that the impairment in learning primarily reflects a reduced ability to

reweight the different cues used for localizing sound. This reweighting has been shown to occur as soon as human listeners experience an earplug in one ear⁷, which could therefore explain why our ArchT ferrets made larger errors than the controls from the first day of monaural occlusion onwards.”

As mentioned in the paper, within-session and between-session learning was explored by including the factors order-of-trials and day-of-plugging in our statistical model, with both contributing to the overall rate of adaptation. Given the relatively small number of trials per session (~100) and the relatively large number (12) of loudspeaker locations over which these trials were spread, it is not possible to break down within-session changes in performance in any more detail other than to say that within-session improvements in performance were observed in both groups. But the ArchT animals perform significantly less well throughout the period of monaural occlusion than the controls.

3. It's unclear how the regression slopes throughout the manuscript were statistically compared. For example, the data in Figure 4 (and the methods section) seem to indicate that a single regression was fit to the mean values in each group. Is this correct? If so, how could the slopes of these two fits be compared with an ANOVA, as suggested on line 162 (and elsewhere throughout the manuscript)? Or were the slopes of these two lines actually compared by looking for a significant interaction of training day and group in an ANCOVA? Please clarify.

All the regression lines were calculated using the data from individual animals, and are not based on the mean values of each group. This is indicated by the degrees of freedom reported in the ANOVA. We thank the reviewer for spotting the ambiguity in the wording used to describe this in the Methods section:

Page 19:

“Rate of adaptation during periods of monaural occlusion was **quantified** by fitting regression lines **for individual animals** to the proportion of correct responses on each day; 95% confidence intervals **around the mean** are shown by shaded areas in the figures. Comparisons of the mean scores obtained at the start and end of each period of monaural occlusion and of the regression line slopes were made using analysis of the variance (ANOVA) after the data were checked for normality by inspection of Q-Q plots and application of the Shapiro-Wilk test, with homoscedasticity tested by Levene's test of equality of variances.”

We were able to compare the slopes using an ANOVA by including dummy variables of 1 or 0 so that the resulting coefficients could be assigned to the different groups and differences between groups determined in the multiple regression model. This approach is equivalent to the method suggested by the reviewer of looking at the interaction term in an ANCOVA.

4. In Figure 6a, the data look bimodal- many animals appear to adapt quickly, in line with the author's hypothesis, but others (at least 3) seem to do poorly, similar to session 1. Could the authors comment on this? How did these three animals perform in the first session?

Two animals showed a slower rate of adaptation during the second earplugging run (filled squares and unfilled triangles), with a third animal showing rapid initial adaptation and a subsequent decline in performance (unfilled squares). We do not have a particular explanation for this beyond individual variability. These 3 animals performed similarly to the others in the first earplugging run (to help readers follow the performance of individual animals, we have now included in Supplementary Figures 4 and 5 the individual animal

symbols used in Figures 4 and 6). Although tempting, we cannot differentiate between slow/bad and quick/good learners. A correlation analysis did not shed any further light on this matter: the correlation coefficients between the performances on the first and last day of the second earplugging run and those achieved on the first and last day of the first plugging run were not statistically significant. Also, when examining the rate of adaptation of the animals in the first earplugging run, the slopes of the individual linear fits for these 3 ferrets (range 0.05-0.074) and the other animals (range 0.03-0.08) completely overlapped. We don't therefore think there is any basis for a bimodal distribution based on learning capacity.

5. The authors state that multiple training sessions were run each day. Were data from each session collapsed into a single day for display and analysis? Did any consolidation or improvement occur between training sessions within a given day?

We tested the ferrets twice daily. In previous studies, we have looked into this matter and consistently found that variations between same-days session are small and definitely smaller than those from consecutive days. We always tested the animals during the wake period, so this observation is consistent with numerous studies showing that sleep plays an important role in perceptual learning. We therefore did collapse the data from sessions carried out on the same day when plotting the change in performance across any period of monaural occlusion. This was also done to facilitate comparison with our previous studies. However, when we submitted the data to our statistical model the factor order-of-trials reflects the order within each session and not the order within a single day – otherwise we would incorrectly exaggerate the contribution of this factor to the model. As described in the Results and shown in Table 2, our GLM showed that localization performance changed significantly with the order of the trials within each testing session and across training days, allowing us to conclude that learning takes place within each session and is then consolidated across sessions.

Minor:

1. Line 153: Typo. I'm assuming the authors meant "P= 0.02 or P = 0.002"?

We meant 0.002; now corrected.

2. Line 355: Typo. "followed" should be "following".

Changed (page 12, para 2).

3. Line 490-1: "Laser stimulation was...paired with... sound presentations, beginning either at the same time." Was there another option? Or were all laser stimulations presented at sound onset?

Sound presentation and laser illumination were always paired at the same time. We have deleted "either" (page 16, para 2).

4. Table 1 and Table 2: The legends indicate that significant values are shown in red text, but there is no red text in either table.

Thank you for spotting this. The revised text was indicated in red and when this text was changed to black in the "clean" version of the manuscript, those significant values shown in red were accidentally changed too. In the present version, and to avoid any confusion, we now

show the significant values in Table 1 and Table 2 in blue and have changed the legend accordingly.

5. While the physiology in Supp. Fig 3 is quite convincing, the merged images are not, and do little for making the case that the recorded neuron actually expressed ArchT. It's clearly the authors decision whether to keep the images in the manuscript or not, but in my opinion, they should either be removed, or replaced with better, more convincing images.

We prefer to keep the images in Supplementary Fig. 3. Whereas panels b and g provide a good indication of the morphology of the recorded neurons filled with biocytin and panels c and h show the expression of ArchT-GFP, without the merged images that show the colocalization of both chromophores in yellow it is impossible to confirm the expression of ArchT-GFP in the labelled neurons. The quality of the images cannot be improved because of the thickness of the slices (400 μm) and the double immunoreactions performed in these thick slices to identify both GFP and biocytin. To help the reader we have added to the figure legend that the colocalization of chromophores results in yellow labeling in the neurons.

Reviewers' Comments:

Reviewer #3:

Remarks to the Author:

The authors have done a thorough job of responding to all of my comments and concerns. I am completely satisfied with their responses.